# Vertical grid refinement for stratocumulus clouds in the radiation scheme of the global climate model ECHAM6.3-HAM2.3-P3

Paolo Pelucchi[1,2], David Neubauer[1], and Ulrike Lohmann[1]

[1]Institute of Atmospheric and Climate Science, ETH Zurich, Zurich, Switzerland
[2]Now at Universitat de València, Valencia, Spain

**Correspondence:** Paolo Pelucchi (paolo.pelucchi@uv.es), David Neubauer (david.neubauer@env.ethz.ch)

**Abstract.** In this study, we implement a vertical grid refinement scheme in the radiation routine of the global aerosol-climate model ECHAM-HAM, aiming to improve the representation of stratocumulus clouds and address the underestimation of their cloud cover. The scheme is based on a reconstruction of the temperature inversion as a physical constraint for the cloud top. On the refined grid, the boundary layer and the free troposphere are separated and the cloud's layer is made thinner. The cloud cover is re-calculated either by conserving the cloud volume (SC-VOLUME) or by using the Sundqvist cloud cover routine on the new grid representation (SC-SUND). In global climate simulations, we find that the SC-VOLUME approach is inadequate, as in most cases there is a mismatch between the layer of the inversion and of the stratocumulus cloud, which prevents its application and is itself likely caused by too-low vertical resolution. Additionally, we find that the occurrence frequency of stratocumulus clouds is underestimated in ECHAM-HAM, limiting a priori the potential benefits of a scheme like SC-VOLUME targeting only cloud amount when present. With the SC-SUND approach, the possibility for new clouds to be formed on the refined grid results in a large increase in mean total cloud cover in stratocumulus regions. In both cases, however, the changes exerted in the radiation routine are too weak to produce a significant improvement of the simulated stratocumulus cloud cover. We investigate and discuss the reasons behind this. The grid refinement scheme could be used more effectively for this purpose if implemented directly in the model's cloud microphysics and cloud cover routines, but other possible ways forward are also discussed.

## 1 Introduction

Stratocumulus (Sc) clouds belong to the low-level stratiform clouds. They occur in many regions and cover large areas of the Earth's surface, but appear most frequently over the oceans. In particular, the subtropical eastern Pacific and Atlantic oceans, west of the continental land masses of North America, South America and southern Africa, experience stratocumulus clouds in excess of 40 % of the time (Wood, 2012), in what are referred to as the semi-permanent subtropical marine stratocumulus sheets. Stratocumulus clouds are of considerable importance to the Earth's radiative budget, as they exert a very strong net negative cloud radiative effect. This is due to a combination of a weak longwave effect due to their low-lying position on the one hand, and of an especially strong reflection of shortwave solar radiation accentuated by their location over dark oceans on the other.

Despite the crucial role of stratocumulus clouds for the climate, their representation in global climate models (GCMs) still has major deficiencies (Boucher et al., 2013). Cloud cover in stratocumulus regions tends to be underestimated (Nam et al., 2012; Neubauer et al., 2014). The representation of stratocumulus clouds is especially challenging due in part to the GCMs' relatively coarse vertical resolution, which degrades the performance of the parametrisations of related processes such as turbulence, convection, microphysics and vertical advection (Yamaguchi et al., 2017). Low vertical resolution can also be the

cause of numerical artefacts such as numerical entrainment (Lenderink and Holtslag, 2000) or spurious radiative-dynamical interactions (Stevens et al., 1999). On a basic level, model gridboxes at the typical level of stratocumulus clouds are generally too thick (a few hundred meters) to resolve the clouds' vertical extent, which can be lower than a hundred meters (Wood, 2012). The resulting overestimation of their vertical extent is associated with an underestimation of their horizontal extent. The models' coarse vertical grid is also not adequate to resolve the temperature profile under which stratocumulus clouds

form, which is characterised by a sharp inversion. Stratocumulus clouds are generally found just below the top of inversion-capped marine boundary layers. The temperature inversion is an essential feature as it suppresses upwelling motion, limiting convection to within the boundary layer and forcing stratocumulus clouds to spread and develop into extended thin sheets. The inversion can be very sharp, attaining a temperature difference of tens of Kelvin in just a few meters vertically (Roach et al., 1982), and as such provides a net separation between the free troposphere and the stratocumulus-topped boundary layer.

Several studies have approached the problem of poor stratocumulus representation via a parametrisation of the planetary boundary layer (PBL) or using vertical grid refinement. An early GCM, which implemented a variable grid level in this context, was the UCLA model (Suarez et al., 1983; Randall and Suarez, 1984). The PBL top was determined prognostically and used as a model interface level. A single model layer was used to represent the whole PBL, and its moist static energy and total water mixing ratio values were used to determine condensation and cloudiness. With this method, the model could cor-

rectly simulate the locations of maximum stratocumulus occurrence, but their incidence was lower than expected. Grenier and Bretherton (2001) developed a moist PBL parametrisation for application to subtropical stratocumulus-capped marine boundary layers, which relies on the assumption that the PBL is topped by an infinitely thin inversion. They present three methods for reconstructing the inversion pressure. Using the inversion pressure to separate the moist PBL and the free troposphere allows continuous evolution of the cloud depth and cloud top location. Grenier and Bretherton (2001) obtained good results with their

scheme and reconstruction in a single column model. However, as they already point out, a scheme with variable grid levels would be difficult to fully implement in a 3D model, where a fixed grid is used for other processes such as horizontal advection. In her PhD thesis, Siegenthaler-Le Drian (2010) applied the diagnostic inversion reconstruction method from Grenier and Bretherton (2001) in the ECHAM-HAM GCM to dynamically refine the vertical resolution in stratocumulus-capped marine boundary layers, adding two new vertical grid levels. However, due to numerical problems, the scheme could not be made fully

interactive in the GCM setup. Also based on Grenier and Bretherton (2001), Bretherton and Park (2009); Park and Bretherton (2009) presented a moist turbulence scheme for the CAM GCM developed at the University of Washington (UW) using the restricted inversion approach. In this case, the model levels are not adapted to match the inversion, but rather a new turbulent mixing and entrainment scheme is applied that includes an explicit entrainment closure. The UW scheme generally improved the simulation of stable boundary layers and hence the climate biases compared to the previous CAM version, in particular

for shortwave cloud radiative forcing. Also stratocumulus cover maxima were better predicted. More recently, Yamaguchi et al. (2017) introduced a framework for enhancing the vertical resolution on which certain physical parametrisations are computed with the aim of improving low cloud representation. The method produced significant improvements in simulations of a drizzling stratocumulus-capped PBL, but at the moment still requires further development and testing.

Other approaches have focused on parametrisation of the cloud cover of stratocumulus clouds and on correcting its bias
due to low vertical resolution using information about the inversion location. Boutle and Morcrette (2010) presented a scheme which separately calculates the cloud area fraction of stratocumulus clouds (as opposed to the volume fraction usually computed in models) from a subgrid interpolation-extrapolation of the vertical temperature profiles, meant to sharpen and better represent the inversion. This fraction is used only for the radiation scheme, but an improvement in the cloud cover and other fields due to internal feedbacks was also observed. In ECHAM-HAM as well, the cloud cover is calculated as the volume
fraction of the gridbox occupied by clouds, and is used as such in the microphysics routine. However the same value is used in the radiation routine as a horizontal area fraction. The underlying assumption reconciling the two interpretations of the model's given cloud cover is that clouds occupy the full vertical extent of a gridbox. This idea results in a misrepresentation of thinner clouds, such as stratocumuli, and their radiative fluxes. Although their in-cloud properties and hence cloud optical depth are correctly estimated, forcing them to occupy the thickness of the model layer results in an underestimation of horizontal extent
- of cloud area fraction - which alters the all-sky radiative flux.

In this study, we develop and implement a new simple parametrisation for stratocumulus cloud cover in ECHAM-HAM. We use the inversion reconstruction from Grenier and Bretherton (2001) to define a refinement of the vertical levels in a way that facilitates a more realistic representation of the horizontal extent of simulated stratocumulus clouds. We do not set out to implement the full PBL parametrisation, and instead focus primarily on the stratocumulus cloud cover, using the exact vertical
location of the reconstructed inversion as a physical constraint for the cloud top. The number of vertical levels does not increase with our approach as the grid boundary atop the cloudy stratocumulus gridbox is shifted to the inversion pressure, producing a thinner gridbox that matches the true vertical extent of the stratocumulus cloud. In one version of our scheme, we rely on conservation of cloud volume to correct the cloud's horizontal extent, i.e. the cloud cover. In the other, we re-calculate the cloud cover using the cover routine applied on the inversion-based refined grid and profile representations. In order to avoid
many numerical problems or difficulties associated with the use of a different grid, we use the new grid and stratocumulus representation only in the radiation scheme of ECHAM-HAM. The radiative effect of stratocumulus clouds is important for climate on a global scale, and hence we hope that the resulting change in radiative transfer and feedbacks can produce an improvement in the simulated stratocumulus clouds overall.

In this article we discuss the implementation and results of our stratocumulus cloud cover parametrisation in ECHAM-HAM.
We describe the new scheme's procedure and details of its implementation in Sect. 2 after giving an overview of the model's current treatment of stratocumulus clouds. In Sect. 3.1, we present the results from a test case in single column model (SCM) mode. In Sect. 3.2 and 3.3, we present the results from global climate simulations and discuss the limitations of the scheme's implementation. Finally, we draw our conclusions in Sect. 4.

## 2 Method

### 2.1 Model description

The work is carried out with the global aerosol-climate model ECHAM-HAM, composed of the general circulation model ECHAM and the aerosol microphysics module HAM, in its version ECHAM(v6.3.0)-HAM(v2.3)-P3. This refers to the latest standard release of ECHAM-HAM (Tegen et al., 2019) used with the P3 microphysics scheme developed by Dietlicher et al. (2018). The horizontal resolution is T63 ($1.875° \times 1.875°$) and the vertical is L47 (47 hybrid sigma-pressure levels). The timestep is 450 s and the radiation routine is ran at 'radiation timesteps' i.e. every 7200 s.

For clouds, ECHAM-HAM-P3 uses a two-moment cloud microphysics scheme with one category for cloud droplets and one for ice, and diagnostic parametrisations for rain. Water vapour, liquid and ice are prognostic variables and the cloud cover is diagnosed. ECHAM-HAM's cloud cover scheme is based on the formulation by Sundqvist et al. (1989). Since absolute humidity in the atmosphere varies on scales smaller than the model's gridboxes, subgrid-scale variations in relative humidity (RH) must be parametrised, in order to achieve the formation of clouds in part of the gridbox. The fraction of a gridbox occupied by clouds is named the fractional cloud cover ($clc$). Given the assumed presence of subgrid variations, clouds must start to form when the gridbox mean RH crosses a threshold value $RH_c$ smaller than the saturation relative humidity, $RH_s = 1$. When the threshold is exceeded, the fractional cloud cover is diagnosed according to Sundqvist et al. (1989):

$$clc = 1 - \sqrt{1 - \frac{RH - RH_c}{RH_s - RH_c}} \ . \tag{1}$$

Under low-level inversions, the formula uses adapted parameters (lower $RH_c$ and $RH_s$) with the aim to facilitate the formation of stratocumulus clouds (Mauritsen et al., 2019).

### 2.2 Scheme description

Because the temperature inversion stops vertical motion at the top of the marine boundary layer, we can equate the inversion with the cloud top, and hence use it to constrain the cloud's position and vertical extent. The cloud cover given by the model is the volume fraction that the cloud occupies in the layer. By conserving the cloud volume and restricting the cloud to be found only below the inversion, we reduce its vertical extent and hence increase the horizontal cloud cover, resulting in a more realistic representation of the stratocumulus clouds. The idea is illustrated with a schematic in Fig. 1. This new grid refinement scheme is called *invgrid*. In the following, we will indicate full-levels (model layers or gridboxes) with an integer index and half-levels (grid boundaries) with half-integers, increasing in the downward direction.

An outline of the method is as follows. First, model columns in which a stratocumulus cloud may be present are identified, and the gridbox layer within which the inversion would be found, named the ambiguous layer, is selected. The exact location (pressure level) of the inversion is diagnosed using the 'reconstructed inversion' method described by Grenier and Bretherton (2001), which assumes a certain sub-grid shape of the temperature profile. The inversion is modelled as a discontinuity in the profile, so that it has a single exact pressure value, usable as the cloud top. Once the inversion pressure is known, the overlying model half-level (representing the cloud top) is shifted down to it, resulting in a thinner lower layer in which the stratocumulus

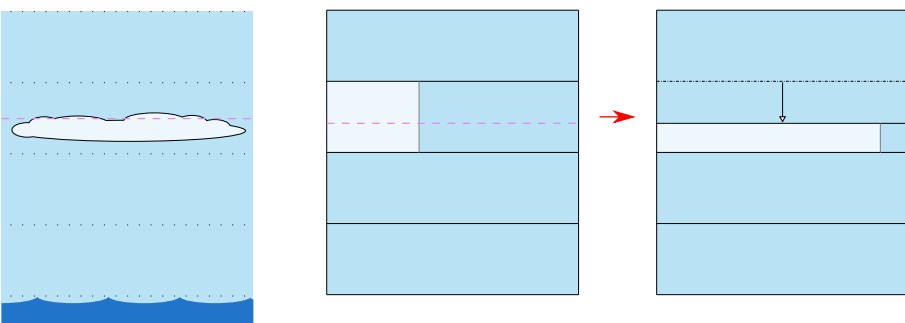

**Figure 1.** Schematic illustrating the idea behind the new stratocumulus representation method. The pink dashed line represents the temperature inversion. On the left is a depiction of the real situation, in the middle its representation in the model's vertical grid, and on the right the same situation on the proposed new vertical grid.

cloud is contained, and a larger but cloud-free above-cloud layer of free tropospheric air. The values of the relevant physical quantities are finally recalculated on this new grid. The new grid boundaries and recalculated quantities are passed to the radiation routine, and the procedure is repeated at every radiation timestep.

With this scheme, the liquid water path (LWP) of the grid-layer is conserved: the layer thickness is reduced and the liquid water mixing ratio is proportionally increased. The in-cloud LWP, which is what is used for cloudy-sky radiative calculations, is reduced by the same proportion that the cloud fraction is increased. As the radiative flux calculation is linear in the cloud fraction but non-linear in the LWP (and hence in cloud optical depth), there can be a difference in radiative fluxes by applying the *invgrid* scheme. We present a demonstration of its effect in Appendix A.

The following sections describe the steps in detail, from the detection of applicable columns to the recalculation of all new-grid quantities. The method to calculate the inversion pressure, described in Sect. 2.2.2, follows closely the 'reconstructed inversion' method developed and described by Grenier and Bretherton (2001). The code to calculate the inversion pressure following this procedure was written by Siegenthaler-Le Drian (2010) for her PhD thesis and hence already available in ECHAM-HAM. A few changes implemented during this study are described.

### 2.2.1 Ambiguous layer selection

The criterion used to select columns in which to apply *invgrid* at each timestep is based on low tropospheric stability (LTS). LTS is a measure defined as the difference in potential temperature between the 700 hPa level and the surface. A strong correlation between LTS and low stratiform cloud cover has been found in observations, especially in the subtropics, as shown by e.g. Klein and Hartmann (1993); Wood and Hartmann (2006). A high LTS is attributable to a strong inversion. Based on the climatology of low stratus cover in Klein and Hartmann (1993), a threshold value of 20 K of LTS is used to select the columns with possible stratocumulus clouds, in which to apply the *invgrid* scheme. This criterion was previously used by Siegenthaler-Le Drian (2010) to select columns in which to activate her stratocumulus-entrainment parametrisation. As a possible alternative, the threshold could also be based on the estimated inversion strength (EIS), which, as a more refined

measure of inversion strength compared to LTS, may be more robust as a predictor of low stratocumulus cloud cover. This is also because, as pointed out by Wood and Bretherton (2006), the relationship between LTS and cloud fraction is not proven
to hold in a warming climate, while the link between EIS and stratocumulus cloud cover is more direct. In the context of this study, the choice of criterion between LTS or EIS is not expected to produce significant differences in the selection of stratocumulus columns, and hence the simpler option is used.

In each identified column, the layer in which to look for and reconstruct the inversion must be selected. It is called the "ambiguous layer" by Grenier and Bretherton (2001) because while in reality it would exhibit a lower cloudy part of boundary
layer air and an upper cloud-free part of free-tropospheric air, within one model gridbox this vertical distinction cannot be resolved. Finding the inversion pressure allows to separate the two parts. To select the ambiguous layer, we first look for the inversion in the model, i.e. the maximum gradient of temperature. This will be found across two grid layers, which may both potentially contain the inversion jump in a sub-grid profile reconstruction. We choose the ambiguous layer as the uppermost of the two possible layer which contains a cloud, defined as non-zero cloud cover and liquid water content (as in the absence
of either of these a cloudy radiative flux is not computed in the model). This selection criterion finds the top of the simulated cloud and hence guarantees that the cloud-rescaling idea would be applicable. If no cloud is present in either of the two possible layers, we use the condition previously used by Siegenthaler-Le Drian (2010): we look at the saturation of an air parcel in an adiabatic ascent from two layers below the inversion, and we choose the ambiguous layer as the layer under the first half-level at which the parcel reaches supersaturation, as it presents the conditions to contain a cloud. This condition operates under the
assumption that the stratocumulus is contained within only one layer, as the cloud top could still be found in the layer above. The scheme allows the possibility to re-attempt the inversion reconstruction calculation one layer above if it fails in the first selected ambiguous layer.

### 2.2.2 Inversion reconstruction

We diagnose the inversion pressure following the method developed and described by Grenier and Bretherton (2001); the
procedure is repeated here for convenience of the reader and to indicate our modifications.

The inversion pressure reconstruction method by Grenier and Bretherton (2001) is based on reconstructing the sub-grid profile of virtual liquid water potential temperature ($\theta_{vl}$) in the ambiguous layer $k$, in which its value $\theta_{vl}^k$ is considered the weighted average of its below-inversion (boundary layer) and above-inversion (free-tropospheric) values. Figure 2 shows a diagram of an example profile, with labelled layer indices.

The virtual liquid water potential temperature is defined as

$$\theta_{vl} = T \left(\frac{p_0}{p}\right)^{\frac{R_d}{c_{pd}}} \left(1 - \frac{L_v}{c_{pd}T}r_l\right) \left(1 + \left(\frac{R_v}{R_d} - 1\right)r_t\right), \tag{2}$$

where $T$ is temperature, $p$ is pressure ($p_0 = 1000$ hPa), $R_d$ and $R_v$ are the dry air and water vapour gas constants, $L_v$ is the latent heat of vaporisation, $r_l$ and $r_t$ are the liquid and total water contents (mass mixing ratios), and $c_{pd}$ is the constant pressure heat capacity of air. $\theta_{vl}$ depends linearly on temperature, and hence exhibits the same vertical profile features (notably
the inversion), with the advantage of being a conserved quantity in a reversible moist adiabatic process, i.e. a process where all

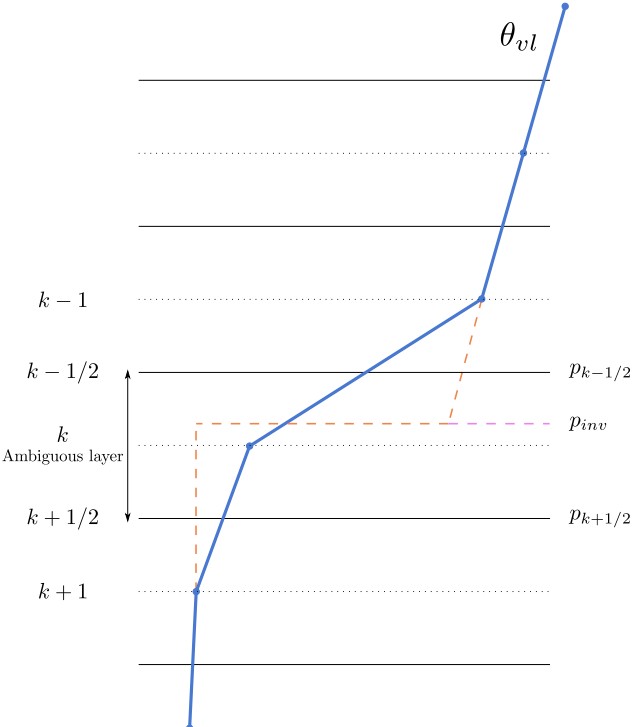

**Figure 2.** Illustration of a $\theta_{vl}$ profile (blue line) and its sub-grid reconstruction (dashed orange line) in the ambiguous layer. The solid and dotted horizontal black lines are the model half-levels and full-levels respectively. The dashed lines extending from levels $k+1$ and $k-1$ represent the assumed sub-grid $\theta_{vl}$ profile within the ambiguous layer, and are obtained as described in Sect. 2.2.2. The discontinuity in the sub-grid profile constitutes the inversion pressure $p_{inv}$.

the condensate remains within the air parcel. The inclusion of the 'potential' and 'liquid water' part (second and third factors on the right hand side of the equation) causes the quantity to be conserved, while the 'virtual' part (fourth factor) allows to use the dry-air equation of state. This makes the quantity advantageous to use in calculations, and hence $\theta_{vl}$ is used for the profile reconstruction.

On a sub-grid scale, within the ambiguous layer, we distinguish between an above-inversion and a below-inversion profile, and assume that $\theta_{vl}$ follows

$$\theta_{vl}(p) = \begin{cases} \theta_{vl}^{k+1}, & p_{k+1/2} > p > p_{inv} \\ \theta_{vl}^{k-1/2} + s\left(p - p_{k-1/2}\right), & p_{inv} > p > p_{k-1/2} \end{cases}. \tag{3}$$

Below the inversion, $\theta_{vl}$ has the same value that it does lower down in level $k+1$, justified by the fact that in the case of a strong inversion, the boundary layer tends to be very well-mixed, in which case $\theta_{vl}$ is constant throughout the well-mixed layer.

Above the inversion, the $\theta_{vl}$ profile is extrapolated down from the overlying level, using the maximum negative gradient with respect to pressure ($s$) chosen from the gradients across half-levels $k-1/2$ or $k-3/2$.

This profile implies that at the inversion pressure $p_{inv}$, $\theta_{vl}$ experiences a discontinuity, where the value jumps from the boundary layer to the free troposphere one, representing the sharp inversion. The inversion pressure is found by requiring conservation of $\theta_{vl}$ within the ambiguous layer, i.e. by requiring that the integral of the sub-grid profile is equal to the original value of $\theta_{vl}$ in the ambiguous layer $k$ ($\theta_{vl}^k$, considered to be the gridbox average):

$$\theta_{vl}^k = \frac{1}{p_{k+1/2} - p_{k-1/2}} \int\limits_{p_{k-1/2}}^{p_{k+1/2}} \theta_{vl}(p) \, dp \tag{4}$$

$$= \frac{1}{p_{k+1/2} - p_{k-1/2}} \left( \int\limits_{p_{k-1/2}}^{p_{inv}} \theta_{vl}^{k-1/2} + s\left(p - p_{k-1/2}\right) dp + \int\limits_{p_{inv}}^{p_{k+1/2}} \theta_{vl}^{k+1} dp \right) \tag{5}$$

In order to solve Eq. (5) for the inversion pressure, we define the above-inversion mass fraction of the ambiguous layer,

$$\mu = \frac{p_{inv} - p_{k-1/2}}{p_{k+1/2} - p_{k-1/2}} \ . \tag{6}$$

Equation (5) can then be turned into a quadratic equation in $\mu$:

$$\frac{1}{2} s \left(p_{k-1/2} - p_{k+1/2}\right) \mu^2 - \left(\theta_{vl}^{k-1/2} - \theta_{vl}^{k+1}\right)\mu + \left(\theta_{vl}^k - \theta_{vl}^{k+1}\right) = 0 \ . \tag{7}$$

The physical solution for $\mu$ is a value between 0 and 1 which, when it exists, can be shown to be the smaller solution of Eq. (7). If and when a physical $\mu$ is found, the inversion pressure $p_{inv}$ is obtained inversely from Eq. (6). In a following step, $p_{inv}$ is used to define the new grid.

A limitation of this method is that it requires a well-mixed $\theta_{vl}$ profile in the PBL to successfully obtain the inversion pressure; specifically, the $\theta_{vl}$ gradient with respect to pressure must be negative both below and above the inversion. While this is a characteristic of stable profiles, we noticed that the method sometimes gave inconsistent results when the profiles slightly deviated from being well-mixed. We included a few minor modifications to the method to allow it to be used in more, although less ideal, situations. For example, we force a small but negative $s$ ($-1 \times 10^{-6} \; \mathrm{K\,Pa^{-1}}$) if the gradient above the inversion is only slightly positive (which would normally be considered unusable). We also attempt to carry out the inversion reconstruction in the upper possible ambiguous layer if it fails in the lower one.

### 2.2.3 Grid refinement

As the new representation is used exclusively in the radiation routine, the grid refinement is applied only in cases where it would make a difference to the radiative transfer calculations, specifically by increasing the cloud cover. Hence, we first check that the ambiguous layer contains a cloud, as this is a necessary condition for the radiation routine to compute a cloudy flux. We also ensure that the gridbox layer would not become thinner than a minimum thickness. The limit is put in place to prevent unphysical situations, such as for example a too-high liquid water mixing ratio or cloud droplet concentration. We choose a threshold of 50 m, as stratocumuli are almost never observed to be thinner (cf. the histogram of observed instantaneous cloud thicknesses in Wood (2012)).

If the conditions are appropriate, we proceed with defining the new refined grid. The half-level above the inversion, the top of the ambiguous layer, is shifted down to the inversion pressure $p_{inv}$. Level $k$ becomes thinner, and will wholly contain the cloud that was originally present in the ambiguous layer. In the case of multi-level clouds, the lower layers are unaffected. Level $k-1$ on the other hand becomes larger, and will represent the first layer of free tropospheric air.

Once the new grid is defined, the variables that need to be passed to the radiation routine are calculated in the new layers using the assumed sub-grid profile, conservation principles, and the notion that the stratocumulus in the new grid is constrained below the inversion. The procedure for each variable is detailed in the following. Superscripts $k$ and $k-1$ refer to variables and layers in the original model grid, i.e. the ambiguous layer and the overlying layer respectively. We use superscripts $kinv$ for the new thinner layer (equivalent to the below-inversion fraction of the ambiguous layer), $abinv$ for the above-inversion fraction of the ambiguous layer (note that this is not a layer in its own right on either grid), and $kinv-1$ for the new larger overlying layer, consisting of layers $abinv$ and $k-1$.

**Water content reconstruction**

The water vapour, liquid water and ice contents are defined as mass mixing ratios ($\mathrm{kg\,kg_{air}^{-1}}$) in the model ($r_v$, $r_l$ and $r_i$ respectively). The total water mixing ratio $r_t$ is the sum of all three individual phases and must be conserved across the affected layers.

For consistency with $\theta_{vl}$, we require that $r_t$ follows the same sub-grid profile in the ambiguous layer, and start by calculating it as

$$r_t^{kinv} = r_t^{k+1} \tag{8}$$

$$r_t^{abinv} = \frac{r_t^k - (1-\mu)r_t^{k+1}}{\mu} \tag{9}$$

where the second equation is a solution to conservation of $r_t$ in layer $k$, given the above-inversion mass fraction $\mu$ of the ambiguous layer. Its value in $kinv-1$ is obtained as a mass-weighted average of $abinv$ and $k-1$:

$$r_t^{kinv-1} = \frac{r_t^{abinv} M^{abinv} + r_t^{k-1} M^{k-1}}{M^{abinv} + M^{k-1}} \tag{10}$$

where $M$ is the air mass of a layer and the denominator is equal to the air mass in $kinv-1$.

Since liquid water and ice are the components that make up the cloud, we restrict them to be found only below the inversion, in the new thinner cloud layer $kinv$. The total liquid and ice mass is conserved, which means the mixing ratio is simply rescaled to the new layer mass:

$$r_{l/i}^{kinv} = r_{l/i}^k \, \frac{M^k}{M^{kinv}} \; . \tag{11}$$

The quantities $r_l$ and $r_i$ are thus assumed to be zero in $abinv$, and for layer $kinv-1$ the values in $k-1$ are rescaled to the larger layer. This recalculation does not change the total in-cloud amounts of liquid and ice water, as the cloud volume is conserved.

The calculation of the water vapour mixing ratio in the new layers $kinv$ and $kinv - 1$ uses the previously calculated reconstructed total water and the inversion-constrained liquid and ice water mixing ratios:

$$r_v = r_t - r_l - r_i \ . \tag{12}$$

We recognise that the method for recalculating the water contents described in this section is not fully consistent. In fact, the total water content is treated as a separate variable (as opposed to using the sum of the individual phases of water) and is reconstructed using the sub-grid profile which depends on the under- and overlying layers; at the same time the liquid and ice contents are taken from the ambiguous layer and simply moved to the below-inversion part. This can lead to an inconsistency in the water vapour content especially in the new cloudy layer $kinv$ (in which the air should be saturated), as it is calculated by subtracting the rescaled liquid and ice contents (from layer $k$) from the total $r_t$ (from layer $k+1$). In $kinv - 1$ the inconsistency is negligible as there should be no liquid or ice water there. We decided to move forward with this method despite this problem because it has the following advantages. With this method, the liquid and ice contents used for the cloud are the ones that are calculated in the cloud microphysics routine, which takes into account fundamental microphysical processes. Also, the resulting total water below the inversion is equal to the one in the layer below, as is characteristic of well-mixed boundary layers. Overall, the method gives reasonable results for $r_v$ above the inversion and for $r_l$ and $r_i$ in the cloudy layer below it, and the total water content as sum of the individual components is indeed conserved. Checks are in place to prevent and fix potential unphysical (negative) values of $r_v$, $r_l$, $r_i$ or $r_t$.

## Temperature

The temperature on the new grid is calculated using energy conservation. First, in $kinv$, $T$ is obtained inversely from $\theta_{vl}^{kinv}$ (Eq. (2)). Then, the internal energy $U$ of the original layers $k$ and $k - 1$ is calculated, along with the internal energy of new layer $kinv$:

$$U^j = \left(c_{pd} + c_{pv}r_v^j + c_{lw}r_l^j + c_{iw}r_i^j\right) M^j T^j \tag{13}$$

where superscript $j$ indicates the layer considered and $c_{pv}$, $c_{lw}$ and $c_{iw}$ are the vapour, liquid water and ice specific heat capacities. Then, for energy conservation over the two layers between the original and new grid, $T^{kinv-1}$ is obtained from

$$T^{kinv-1} = \frac{U^k + U^{k-1} - U^{kinv}}{\left(c_{pd} + c_{pv}r_v^{kinv-1} + c_{lw}r_l^{kinv-1} + c_{iw}r_i^{kinv-1}\right) M^{kinv-1}} \ . \tag{14}$$

### Further cloud variables

Similar to $r_l$ and $r_i$, we confine all cloud variables of the ambiguous layer $k$ to the new thinner layer $kinv$, which is capped by the inversion. The recalculation invokes conservation of cloud volume for cloud cover ($clc$), and particle number for cloud droplet and ice crystal number concentrations ($n_{cd}$, $n_{ic}$). The variables are simply scaled to the new layer thickness $Z$, essen-

tially 'squeezed' under the inversion:

$$n_{cd/ic}^{kinv} = n_{cd/ic}^{k} \frac{Z^k}{Z^{kinv}} \tag{15}$$

$$clc^{kinv} = clc^{k} \frac{Z^k}{Z^{kinv}} \tag{16}$$

The cloud cover is of course constrained not to exceed 100 %.

The new half- and full-level pressures (grid boundaries) and all the recalculated new-grid variables are finally passed to the radiation routine.

Note that the aerosol tracers are not regridded in *invgrid*. The slight alteration of the *invgrid* radiative effect due to this is of secondary importance to the effects caused by changes in cloud cover and condensate. The more important aerosol effects on
clouds and how they are parametrised are kept the same.

## 2.3   Model versions

The model versions that were used to perform the simulations discussed in the next sections are presented in the following. In addition to the reference model version (REF), two versions implementing *invgrid* were used: one which rescales cloud cover based on cloud volume conservation, as described above, (SC-VOLUME); one which re-calculates the cloud cover on
the refined grid by re-running the model's Sundqvist cloud cover scheme (SC-SUND). Another simple scheme (SC-MAX) was used to test and provide an understanding of the potential and limitations of the different *invgrid* versions.

### 2.3.1   REF

The model version ECHAM(v6.3)-HAM(v2.3)-P3 is used as the base model version and referred to as REF (Dietlicher et al., 2018; Tegen et al., 2019). Simulations conducted with REF illustrate the baseline performance of the model and provide the
reference to which simulations conducted with the new schemes developed in this thesis are compared. A brief description of REF and its relevant schemes is given in Sect. 2.1 of this chapter.

### 2.3.2   SC-VOLUME

In the SC-VOLUME model version, the *invgrid* scheme described in Sect. 2.2 is fully implemented in the model. The calculation of the inversion pressure as in Sect. 2.2.2 is performed at every timestep before the radiation routine for diagnostic
reasons. At radiation timesteps, the value is used to refine the vertical grid; physical variables are recalculated as described in Sect. 2.2.3, with the stratocumulus cloud cover calculation being based on cloud volume conservation. These are passed to the radiation routine.

### 2.3.3 SC-SUND

In the SC-SUND model version, after executing the *invgrid* grid refinement, the stratocumulus cloud cover is calculated by
running the model's Sundqvist cloud cover scheme. This is done regardless of the original cloud cover. The goal is in particular to address cases in which, on the original grid, no cloud is present in the ambiguous layer. This could be due to the ambiguous layer's water vapour mixing ratio being an average between dry tropospheric air and moist boundary layer air, which may cause the gridbox average relative humidity to be too low to reach the threshold for forming cloud cover according to Eq. (1). With the new grid's reconstruction, the two different air masses are separated, which may allow a cloud to form in the new
thinner layer, now made up exclusively of boundary layer air and hence presumably having a higher relative humidity. This would be valuable because it would lead to a better representation of stratocumulus clouds in layers in which the method of SC-VOLUME could not be applied due to the lack of a cloud in the model in the first place. This method makes use of the refined grid and recalculated profiles of water content and temperature, but the cloud volume is not necessarily conserved as the cloud cover is recomputed with the cloud cover scheme. The procedure is only applied if the layer in which a new cloud
cover is calculated already contains liquid water (or cloud ice), to ensure the presence of a 'real' cloud (having cloud cover and water condensate), since the Sundqvist cloud cover scheme itself does not consider or affect the presence of condensate. The new cloud cover representation is only used in the radiation routine.

### 2.3.4 SC-MAX

The SC-MAX model version was designed to investigate the maximum possible effect of a scheme that increases the cloud
cover of existing stratocumuli, such as in SC-VOLUME. This is done by always increasing the cloud cover to 100 % in model layers where a stratocumulus cloud is identified. The cloud cover increase is applied in the same cases in which SC-VOLUME's cloud rescaling would be, i.e. when the identified ambiguous layer contains a cloud, but also when the ambiguous layer contains no cloud but another layer, at most two levels below it, does. We still consider the latter case as a stratocumulus cloud. The cloud cover of the first (uppermost) cloudy model layer below the inversion is set to 100 %. The modified cloud cover is passed
to the radiation routine.

### 2.4 Experiment description

For each model version, we performed a 15-year-long (2000–2014) global climate simulation with prescribed AMIP sea-surface temperatures (PCMDI, 2018) to evaluate the stratocumulus cloud representation. We used the standard ECHAM-HAM T63/L47 spatial resolution and $450\,\mathrm{s}$ (7.5 minute) timestep. The data from the *invgrid* routine is sampled at radiation timesteps,
i.e. every 2 hours. As an observational reference for total cloud cover, we used Cloud-Aerosol Lidar and Infrared Pathfinder Satellite Observations (Calipso) data from the GCM-Oriented Calipso Cloud Product (GOCCP) dataset (Chepfer et al., 2010).

## 3 Results and discussion

### 3.1 Single Column Model

We first tested *invgrid*'s inversion reconstruction and grid refinement in ECHAM-HAM's single column model (SCM) mode (Dietlicher et al., 2018). Using the SCM allows us to closely observe the evolution of the vertical profiles and how *invgrid* responds to them. Additionally, the possibility to use observational forcings for the SCM is a method to test the model's representation of real situations and to generally validate the reconstructions of the new scheme. The validation in the SCM was carried out using a forcing derived from observations made during the EPIC campaign (Bretherton et al., 2004), specifically from a segment between 16–22 October 2001 in the southeastern Pacific, where the vertical structure of the boundary layer capped by a persistent stratocumulus cloud was observed using radiosondes and remote sensing (Bretherton, 2005). The EPIC campaign also provided observations of the cloud top and base in this period, obtained with cloud radar and ceilometer respectively (Caldwell et al., 2005), which are used to validate the found inversion heights.

Figure 3 shows the evolution of the cloud top and base over the 6-day period of the EPIC campaign. The cloud top follows the PBL's diurnal cycle, rising during the night due to longwave cloud top cooling driving entrainment and sinking during the day due to absorption of solar radiation suppressing entrainment. The reconstructed inversion generally captures this diurnal cycle. While the exact height of the inversion is at times overestimated (days 1, 4), most of the time it matches the observed cloud top quite well, especially on days 2, 3 and 6. The occasional sudden jumps of the inversion pressure (e.g. between days 1 and 2) occur due to the selection criterion for the ambiguous layer depending on the maximum gradient of $\theta_{vl}$, whose level can change suddenly when the inversion is not very sharp. Finding a criterion which could address this undesirable issue without loss of generality proved difficult. We also calculated the lifting condensation level (LCL) for an air parcel rising from the surface to attempt to estimate the cloud base, but the results exhibited large oscillations and did not match the cloud base most of the time, rendering the LCL diagnostic unsuitable as a proxy for cloud base. A reconstruction of the cloud base would be beneficial to the scheme to complement the constraint of the cloud's extent from above with a constraint from below, resulting in a further improved representation, but the development of a method for accurately diagnosing the cloud base of stratocumulus clouds is outside the scope of this study. In comparison to the inversion reconstruction performed by Siegenthaler-Le Drian (2010), who also tested it in the SCM with the EPIC data, with the modifications added in our method the inversion is found also in cases where the method previously failed, for example at the start of the first day or during day 6.

For further validation we also ran the EPIC SCM experiment with different relaxation timescales and perturbed initial conditions; the results are presented in Appendix B. Overall, the inversion reconstruction method gives good results for finding the location of the stratocumulus cloud top.

We show some example vertical profiles which occurred during EPIC to illustrate the effect of the grid refinement on temperature, total water mixing ratio and cloud cover in Fig. 4. The inversion in the physical quantities is sharper, and demonstrates the better separation between the PBL and the free troposphere on the refined grid. The cloud cover increases by 6 percentage points - or by almost 30 % - as a result of the lower vertical extent of the layer.

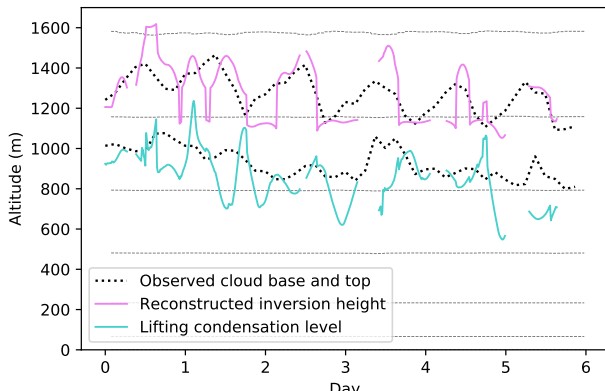

**Figure 3.** Observed cloud base and top during the stratocumulus segment of the EPIC campaign (Bretherton, 2005), and reconstructed inversion and LCL height. The grey dashed horizontal lines are model half-levels in ECHAM-HAM. The time series starts on 15 October 2001, 18:00 local time. The cloud and its boundaries as represented by the model can be seen in Fig. 5 (REF).

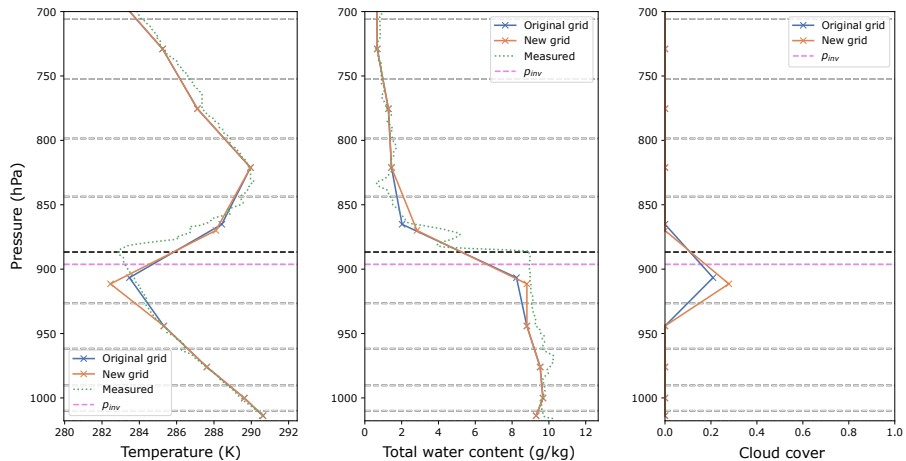

**Figure 4.** Example vertical profiles of the EPIC SCM experiment obtained with the original and new (refined) model grid and observed profiles at the end of day 5 (hour 118). The grey dashed lines represent half-levels common to both grids. The black dashed line represents the top of the ambiguous layer on the original grid; it is shifted to the inversion pressure (magenta dashed line) for the refined grid.

The EPIC SCM experiment was simulated with the REF, SC-VOLUME and SC-SUND model set-ups; Fig. 5 shows the cloud cover below 800 hPa for each experiment. Only the cloud cover belonging to layers which also have a non-zero liquid water content is shown. In the REF simulation, the cloud is mostly contained within one layer in the model. In fact, it is found most often in the layer below the one containing the inversion, and hence in which cloud top was predicted (and observed) to be. In these situations, with SC-VOLUME the cloud rescaling cannot be applied, as the layer containing the inversion contains no cloud, and hence the reconstruction would not make a difference to the radiation routine. The three times where the

model's cloud extends into the upper layer, the *invgrid* scheme is applied effectively, reducing the thickness of the top cloudy layer following the inversion and hence obtaining a more realistic depiction. The SC-SUND scheme version was developed in response to the issue described above: it uses a better water profile representation by applying the refined grid also in the cloud cover routine, so that possibly a new cloud can be formed right below the inversion that is missing when using the original grid.

In the SC-SUND simulation, a new cloud is formed in a few cases in the upper layer (days 1, 2, 4) and once in the central layer at the end of day 5. The scheme actually simulates a new cloud cover more frequently, but the lack of water condensate in the inversion layer limits the number of 'valid' instances. A more ideal representation of the water content in addition to $clc$ would be obtained if the grid refinement method were used in the microphysics routines too. Such an implementation comes with the aforementioned challenges that our more limited usage of the new scheme in the radiation routine only aimed to avoid.

In our analysis of the global simulations we also investigate the frequency of situations such as those observed in the EPIC simulations, in which the model's simulated cloud is in the layer below where we expect to find it via the inversion reconstruction.

## 3.2 Global Climate Simulations

After demonstrating the desired functioning of *invgrid* in the SCM, we studied its effect on the stratocumulus cloud cover in
global climate simulations. We focus on three subtropical stratocumulus regions, which are known to exhibit semi-permanent marine stratocumulus sheets, namely the oceans just west of North America (NAM), South America (SAM) and southern Africa (AFR). We also look at an Arctic region, over the Barents sea (BAR). The regional averages cited in the text are defined over the areas highlighted in Fig. 6a.

The reference model version REF generally underestimates cloud cover in the subtropical stratocumulus regions, as shown
in Fig. 6b in a comparison to the Calipso-GOCCP satellite climatology (Chepfer et al., 2010). The cloud cover difference exhibits a similar pattern in all three regions: compared to observations, cloud cover is actually overestimated along the coast, such that the overall underestimation results from large areas of lower cloud cover further offshore. In the Arctic, total cloud cover is instead overestimated by the model.

The results from the simulations with the modified schemes are shown in Fig. 6c-h, with on the right hand side the annual
mean simulated total cloud cover, and on the left hand side the total cloud cover change experienced by the radiation routine, i.e. the difference between after and before the application of the *invgrid* scheme. The total cloud cover in the simulations can change when changes by the invgrid scheme on cloud radiative effects feed back on the clouds, e.g. by increased turbulence through stronger cloud top cooling. Regional averages are reported in Table 1.

In the SC-VOLUME simulation, the increase in total cloud cover caused by *invgrid* and seen by radiation in the annual
mean is extremely small in stratocumulus regions, reaching at most 1 percentage point (pp) in the Arctic where it is most marked. As the changes in the radiation routine are small, the change induced to the simulated cloud cover due to internal climate feedbacks is also very small. A simple two-sided t-test using the annual means showed that the results do not differ from REF in a statistically significant manner; they also do not exhibit an explicable pattern (Fig. 6d). The changes in cloud

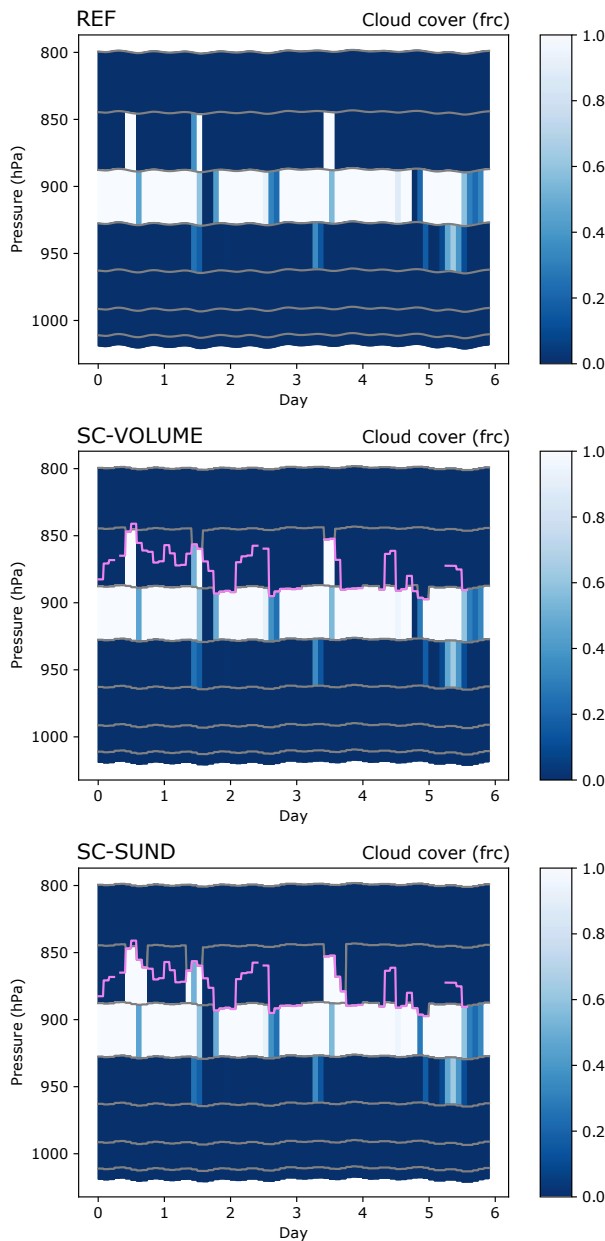

**Figure 5.** Cloud cover fraction of clouds in the EPIC SCM simulations in REF, SC-VOLUME and SC-SUND. The grey lines represent the model half-levels, the magenta line is the reconstructed inversion pressure. The inversion's evolution in time appears more step-like than in Fig. 3 because the refined grid is only applied for the radiation routine at radiation timesteps (every 7200 s).

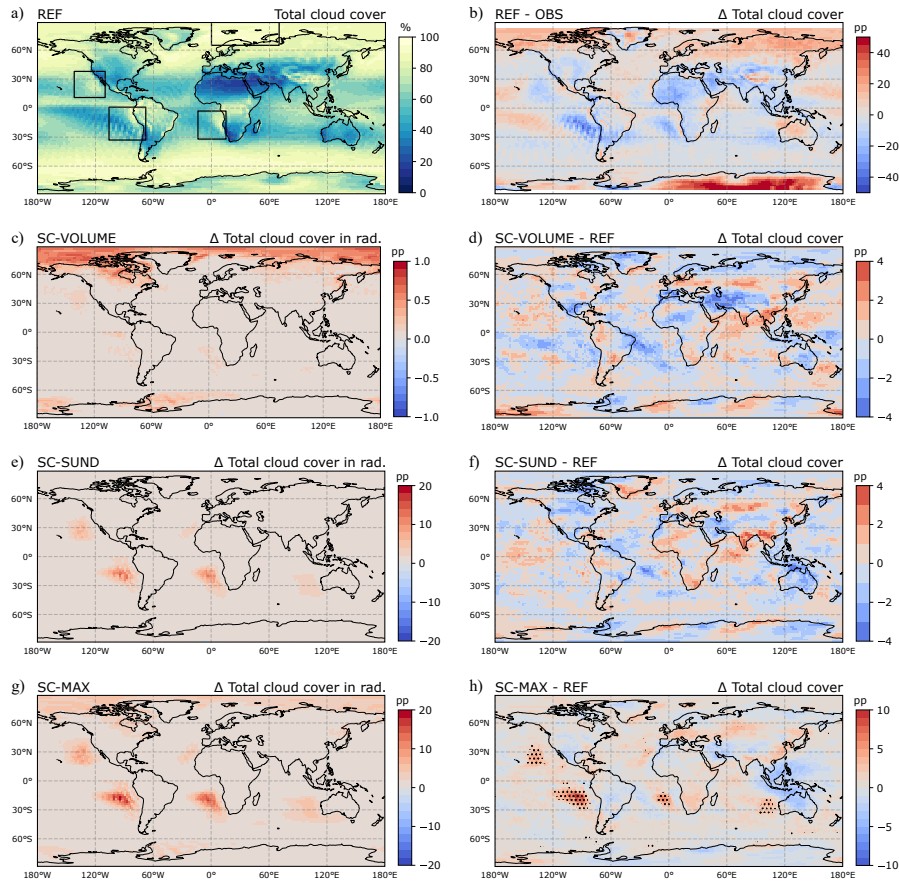

**Figure 6.** Total cloud cover results from 15-year free global climate simulations. Reference results (REF): (a) total cloud cover, (b) difference with Calipso climatology. Results with SC-VOLUME, SC-SUND, SC-MAX: (c)–(h) on the left, total cloud cover increase exerted in the radiation routine; on the right, change in simulated total cloud cover compared to REF (stippling indicates statistically significant differences at the 95 % significance level; the false discovery rate is controlled following Wilks (2016)).

The regions highlighted in Fig. 6a are defined as follows: NAM: $38°$ N–$11°$ N, $142°$ W–$110°$ W; SAM: $1°$ N–$33°$ S, $106°$ W–$68°$ W; AFR: $3°$ S–$32°$ S, $14°$ W–$15°$ E; BAR: $90°$ N–$65°$ N, $0°$ E–$70°$ E.

radiative effects produced with SC-VOLUME were much weaker than we had initially expected (not shown), and in Sect. 3.3
we investigate the factors that limit the effectiveness of the SC-VOLUME method in global simulations.

    In the SC-SUND simulation, the possibility to form new clouds on the refined grid gives the potential to produce a larger mean cloud cover increase than with SC-VOLUME. This is in fact the case in the radiation routine (Fig. 6e): as intended, the subtropical stratocumulus regions exhibit large increases (up to 15 pp) in the annual mean total cloud cover. The most affected areas are located away from the continental coasts, i.e. in the regions where ECHAM-HAM most underestimates cloud cover,
showing that SC-SUND can accurately address the problem. As for the change induced in the simulated total cloud cover (Fig. 6f), the difference to REF is also small (on the same order of magnitude as with SC-VOLUME), although the spatial

**Table 1.** Annual mean total cloud cover and differences between simulations, as global and regional averages. An asterisk denotes statistically significant differences at the 95 % significance level. "Δ seen by rad. " indicates the change in total cloud cover produced with *invgrid*, which is then applied only in the radiation routine.

| Total cloud cover | Global | NAM | SAM | AFR | BAR |
|---:|:---:|:---:|:---:|:---:|:---:|
| Calipso (%) | 67.2 | 69.1 | 71.7 | 66.5 | 82.6 |
| REF (%) | 66.4 | 63.0 | 58.3 | 58.2 | 89.6 |
| REF minus Calipso (pp) | -0.9 | -6.0 | -13.5 | -8.4 | +7.0 |
| SC-VOLUME minus REF (pp) | -0.12 | -0.39 | -0.60* | -0.48 | -0.70* |
| SC-VOLUME Δ seen by rad. (pp) | +0.04 | +0.05 | +0.06 | +0.06 | +0.18 |
| SC-SUND minus REF (pp) | -0.10 | +0.12 | +0.08 | 0.00 | -0.46 |
| SC-SUND Δ seen by rad. (pp) | +0.57 | +2.13 | +3.01 | +3.63 | +0.47 |
| SC-MAX minus REF (pp) | +0.26* | +1.44* | +1.72* | +1.36* | +0.01 |
| SC-MAX Δ seen by rad. (pp) | +0.99 | +3.23 | +4.37 | +5.19 | +1.29 |

patterns seem to indicate a slight reduction of the model bias in subtropical stratocumulus regions. The stronger cloud top radiative cooling could favour convection bringing moisture into the cloud from the surface, increasing stratocumulus cloud longevity in a positive feedback loop affecting the simulated cloud cover. However, the difference with REF was found to be
not statistically significant with SC-SUND as well.

With both set-ups, the change exterted is too small to cause significant changes in the simulated total cloud cover. At the same time, the results indicate that, in terms of the initial changes produced in the radiation routine, SC-SUND is more effective than SC-VOLUME at increasing cloud cover in the annual mean. This suggests that the model's bias is less due to an underestimation of cloud extent in individual instances, which SC-VOLUME is designed to address, and more to a negative
bias in the frequency of stratocumulus cloud formation, which can be addressed by SC-SUND due to its re-evaluation of cloud cover on the refined grid. Other factors hindering the suitability of SC-VOLUME could also be at play, and they are considered in the following Section.

### 3.3 Further analysis and scheme limitations

#### 3.3.1 Scheme usage frequency in SC-VOLUME

For SC-VOLUME's cloud cover reconstruction to produce a significant effect in global simulations, it must be applied frequently in practice. The *invgrid* scheme requires a series of conditions to be met in order to be applied and rescale the ambiguous layer's cloud cover. The occurrence frequency of these conditions in the SC-VOLUME simulation is reported in Table 2.

First of all, the scheme must find and successfully reconstruct a temperature inversion. The associated conditions and calcu-
lations, described in Sections 2.2.1 and 2.2.2, result in the inversion being found very frequently in the stratocumulus regions,

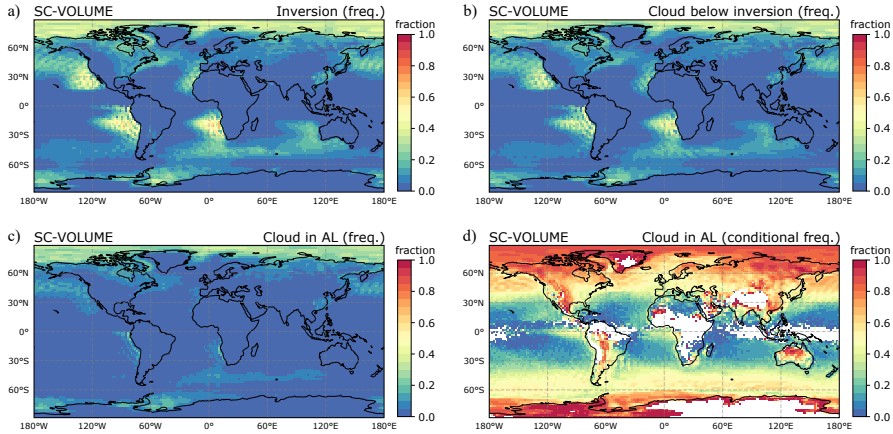

**Figure 7.** Frequency of occurrence of various conditions related to *invgrid* in the SC-VOLUME simulation: (a) an inversion is found, (b) a cloud is present below the inversion, (c) a cloud is present in the ambiguous layer, (d) given that there is a cloud below the inversion, the cloud is in the ambiguous layer (conditional).

upwards of 70 % in some columns (Figure 7a). In most of these cases, a stratocumulus cloud, defined as a cloudy layer at or below the inversion, is also present (Figure 7b). The occurrence frequency of these identified stratocumulus clouds is lower than in reality, where it is around 46 % annually in the relevant regions according to the ship-based observational climatology (1954–1997) by Hahn and Warren (2007). This represents a deficiency of the model and a limitation to the SC-VOLUME

scheme's aptness to correct the cloud cover bias. The method can only target errors in cloud cover amount when a cloud is present, so a model bias in cloud occurrence frequency puts an a priori limit to its possible benefit. The practical applicability of SC-VOLUME's cloud reconstruction method is even more starkly reduced by the subsequent necessary condition, that the cloud (or at least its upper part) must be found in the same model layer as the inversion. As indicated in Sect. 3.1 and quantified in Fig. 7c, this condition is in fact very rare and occurs in much more limited areas than those in which stratocumulus clouds

are identified, concentrated in close proximity of the coasts. Figure 7d shows the conditional probability of the stratocumulus cloud being found in the ambiguous layer, given that a cloud is present below the inversion. This probability decreases with the distance from the coast in the subtropical marine stratocumulus regions, where overall it is less than 25 %. The rest of the time, the cloud is at a lower level than the inversion. The conditional probability is instead very high in higher latitudes. This is likely the result of the different meteorological conditions - due to lower temperatures and the presence of ice, the model's RH

requirement for cloud cover formation is lower and easier to reach. In addition, the PBL is typically shallower in the Arctic, and as the model's vertical resolution is higher closer to the surface, its vertical structure is better resolved and can more easily form clouds at the right level.

     The results indicate that there is a prevalent mismatch between the layer where the inversion is found, which is where the cloud top is expected to be, and where the model in fact forms the cloud, in the layer below the inversion. In these cases,

the idea of 'squeezing' the existing cloud under the inversion cannot be used, and hence this discrepancy between predicted

**Table 2.** Global and regional average occurrence frequency in the SC-VOLUME simulation of finding an inversion, finding an inversion with an underlying cloud (identified stratocumulus clouds), finding a cloud in the ambiguous layer (AL); and the conditional occurrence frequency of a cloud in the ambiguous layer, given that a cloud is present below the inversion. Also included for comparison are the average frequency of occurrence of stratocumulus clouds from the Hahn and Warren (2007) surface-based observational cloud climatology, averaged over the stratocumulus regions defined in this study.

| | Global | NAM | SAM | AFR | BAR |
|---|---|---|---|---|---|
| Inversion found (%) | 6.7 | 20.9 | 21.5 | 26.6 | 16.1 |
| Identified Sc cloud (%) | 5.6 | 17.3 | 16.5 | 20.8 | 14.9 |
| Cloud in AL (%) | 2.5 | 4.0 | 3.5 | 4.0 | 11.1 |
| Conditional cloud in AL (%) | 37.4 | 23.6 | 24.0 | 20.9 | 69.8 |
| | Global (oceans only) | | Stratocumulus regions | | |
| Obs. Sc occurrence frequency (%) | 31 | | 46 | | |

and effective location of the cloud in the model greatly reduces the applicability of SC-VOLUME's method, especially in the subtropics.

As the SC-SUND simulations in Sect. 3.2 indicate, the origin of the discrepancy may lie in the nature of the ambiguous layer itself. As it is located across the inversion, variables such as temperature and water vapour concentration would in reality be very different between the bottom and the top of the layer. The ambiguous layer's values hence represent an average of the cold and possibly moist PBL air at the bottom and dry and warm free-tropospheric air at the top. Then, depending on the proportion of boundary layer versus free tropospheric air (i.e. depending on where the inversion lies within the layer), the gridbox mean saturation may or may not be sufficient to form a cloud (see Sect. 2.1). Cloud formation is favoured in high RH conditions of the PBL air, and therefore will be unlikely in the ambiguous layer especially when the inversion is close to its bottom. The layer below the ambiguous layer, being fully located inside the PBL, is instead much more likely to present the conditions appropriate to form a cloud in the model. Thus the stratocumulus cloud is most often found in the layer below the ambiguous layer, rather than in the ambiguous layer itself. The problem we identified of misrepresentation of the cloud's vertical location seems to be a result of poor vertical resolution, just like the underestimation of stratocumulus cloud cover due to exaggeration of their vertical extent. Our scheme aimed to correct the latter, but doing so as we envisioned is difficult without also addressing the former.

### 3.3.2 Maximum cloud cover improvement with SC-VOLUME

In addition to only being used in a small fraction of stratocumulus cases, we found that SC-VOLUME's cloud reconstruction does not tend to increase cloud cover very much in the layers in which it is used. Figure 8a shows the mean cloud cover in the ambiguous layer when it contains a cloud. In the stratocumulus regions, close to the coasts the ambiguous layer cloud cover is already very high on average, and hence cannot be increased much further, but farther offshore it decreases as low as 40

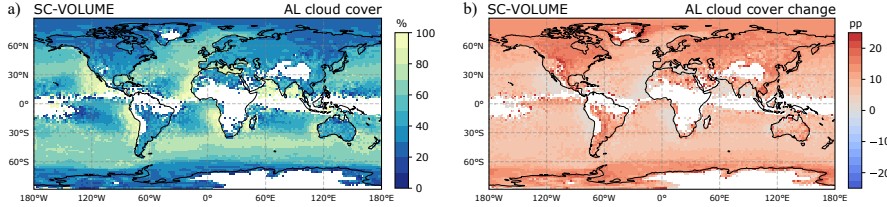

**Figure 8.** Mean (a) cloud cover and (b) cloud cover change in the ambiguous layer with SC-VOLUME, conditionally sampling cases in which the ambiguous layer contained a cloud.

**Table 3.** Global and regional averages of mean ambiguous layer cloud cover ($clc^{kinv}$) and its increase in SC-VOLUME, conditionally sampling cases in which the ambiguous layer contained a cloud.

|  | Global | NAM | SAM | AFR | BAR |
|---|---|---|---|---|---|
| Original $clc^{kinv}$ (%) | 48.7 | 76.3 | 72.4 | 76.5 | 35.4 |
| SC-VOLUME $\Delta clc^{kinv}$ (pp) | +0.5 | +3.7 | +4.4 | +3.4 | +13.6 |

%. However, the mean increase produced there is less than 10 pp (Fig. 8b). A probable reason for this is that, when inversion and cloud layer match, the inversion is likely to be high within the layer (as it is the associated higher proportion of PBL in the layer that allowed the formation of a cloud). Hence, the refined layer is not much thinner than the original one, and a volume-conservation-based reconstruction of the cloud cover does not increase it very much. This demonstrates again how the

SC-VOLUME method for cloud cover reconstruction is limited by the same biases of the original vertical representation that *invgrid* aims to correct. While the grid refinement can improve the vertical representation, basing the new cloud cover on the flawed original cloud cover gives poor results.

To assess the maximum effect that a scheme such as *invgrid* in SC-VOLUME, increasing the cloud cover for existing stratocumulus clouds, could cause, we performed the SC-MAX experiment. In this simulation, the cloud cover of identified

stratocumulus clouds, i.e. the first cloudy layer at or below the inversion level, is set to 100 %. The SC-MAX method is applied also to those stratocumulus situations that SC-VOLUME could not affect (in which the cloud and the inversion are not in the same layer). Hence, the SC-MAX method exerts the maximum possible stratocumulus cloud cover increase.

The annual mean total cloud cover difference that is produced for the radiation routine with SC-MAX is very large, as can be seen in Figure 6g. Further, in this case the changes exerted propagate through feedbacks much more evidently, and can be

clearly observed in the model's simulated total cloud cover. When comparing it to REF (Fig. 6h), the increase exhibited in the subtropical stratocumulus regions is significant. However, the model's bias compared to observations is still far from being completely corrected, as the average underestimation in the South American region, which experienced the most improvement, is still -11.7 pp with SC-MAX as opposed to -13.5 pp in REF.

The SC-MAX experiment demonstrates that a stratocumulus cloud cover scheme applied only in the radiation routine can

have a positive effect on the model via feedbacks, but, in the case of ECHAM-HAM, it is not sufficient to fully close the gap between the simulated and observed cloud cover. Even the cloud cover seen by the radiation routine is still underestimated in the stratocumulus regions compared to the observed climatology. This experiment further confirms that ECHAM-HAM's cloud cover bias is caused also by a lack of stratocumulus clouds in the first place. A scheme such as SC-VOLUME can only correct the cloud cover when a cloud is already present, and as such it has limited effectiveness when the main model bias is the

frequency of cloud occurrence. The implementation of a scheme affecting only existing clouds, such as SC-VOLUME, would need to be complemented by improvements in other parametrisation schemes to increase the occurrence of stratocumulus clouds as well. It would be better suited for models that correctly simulate stratocumulus frequency but have too low cloud cover when present.

The SC-SUND scheme presents a possible improvement to this, as it can be applied even in columns with no below-inversion

cloud at all, with the possibility to form a new cloud there.

### 3.3.3   New clouds in SC-SUND

The SC-SUND scheme has the potential to address both the issues identified in the previous sections in the SC-VOLUME set-up, namely its inability to address cases in which the stratocumulus is below the inversion layer and the scarcity of simulated stratocumulus in the model. It can form a 'new cloud' when on the new grid the Sundqvist scheme diagnoses positive cloud

cover in a layer which previously had zero cloud cover but some condensate.

Figure 9a shows the frequency of occurrence of a cloud in the ambiguous layer after SC-SUND's reconstruction, and Fig. 9b its increase compared to the original representation. The SC-SUND scheme forms new clouds up to 30 % of the time in the southern marine subtropical stratocumulus regions, also significantly extending over the ocean the area over which the condition occurs. This demonstrates that the separation of PBL and free troposphere achieved with the refined grid is very

effective, and allows clouds that could not be formed previously in the coarser gridbox to be 'revealed' on a re-execution of the cloud cover scheme. The overall effect that a newly formed cloud in the ambiguous layer has on total cloud cover depends on the presence of a cloud in other layers: in a previously cloud-free column a new cloud increases the total cloud cover in general more significantly than in a column already containing a cloud.

The total cloud cover change experienced in the annual mean by the radiation routine in the SC-SUND simulation is quite

large, reaching up to 15 pp in some columns and affecting extended areas (Fig. 6e). It is of comparable magnitude to that in the SC-MAX simulation (Fig. 6g). However, it is interesting to note that the effect that is then produced on the simulated total cloud cover is much less marked with SC-SUND than with SC-MAX (see Fig. 6f and 6h). In fact, the process by which the annual mean total cloud cover seen by the radiation routine is increased is different in the two set-ups. In SC-MAX, the annual mean increases because all simulated stratocumulus clouds become 100 % covering, but their number remains the same. In

SC-SUND, it increases because the scheme can form new clouds and hence stratocumulus clouds appear more frequently, with their coverage being calculated as usual. Thus, in SC-MAX, sunlight is scattered back to space almost fully in all situations with a stratocumulus cloud, which can have a very drastic effect on the radiation balance in stratocumulus regions; on the other

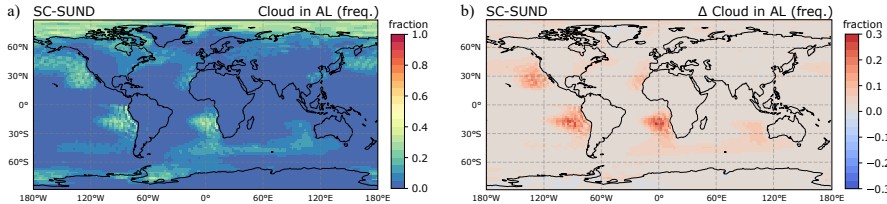

**Figure 9.** Presence of a cloud in the ambiguous layer in the SC-SUND simulation: (a) frequency (cf. Fig. 7c), (b) frequency increase compared to before application of SC-SUND.

hand in SC-SUND the increase in shortwave scattering in stratocumulus regions is more evenly distributed over time and may hence produce a more moderate effect on the meteorological conditions in those regions.

The reason why the effect produced in SC-SUND is weaker may be because the new clouds occurring in the scheme, although they have a more realistic cloud cover, are likely to have a too-low liquid or ice content. Their liquid or ice content comes from the condensation or deposition computed using the original grid's gridbox mean RH, i.e. at low supersaturation, or from transport - both resulting in low amounts. This is a disadvantage of SC-SUND, as to have realistic liquid or ice content the grid refinement should be applied in the cloud microphysics scheme too.

To test this explanation, we evaluated the difference in shortwave (SW) cloud radiative effect (CRE) between the REF and SC-SUND simulations, which gives us information about whether the new clouds formed in SC-SUND are radiatively different from the 'original' ones in REF. As the CRE is the difference between all-sky and clear sky radiative fluxes, its magnitude can change based on changes in cloud cover, cloud occurrence frequency or cloud optical thickness. A more negative CRE can be expected in stratocumulus regions in SC-SUND as both cloud cover and cloud frequency increased in the radiation routine,

as long as the cloud optical thickness is large. Figure 10 shows the difference in mean SW CRE between the REF and SC-SUND simulations. Stippling shows regions where the difference is statistically significant at the 95 % significance level. In stratocumulus regions, where an important increase in mean cloud cover was simulated (Fig. 6e), the SW CRE difference is still not significant. This can only be explained if the mean optical thickness of newly formed clouds, primarily responsible for the mean cloud cover increase in the radiation scheme in SC-SUND, was abnormally low and close to zero. This provides an

explanation as to why application of SC-SUND only in the radiation routine did not have the desired effect despite the large increase in cloud cover and occurrence - as suggested, the newly formed clouds are devoid of significant cloud condensate and hence are not radiatively active. If the new clouds were comparable to the pre-existing clouds in terms of water condensate, a strong radiative change would be observed, leading to favourable feedbacks, like in SC-MAX.

These results indicate that the cloud cover improvement obtained in the radiation routine thanks to the implementation of

*invgrid* is 'lost', as the radiative impact is too weak for the changes to be propagated to the simulated climate, due to a too-low water content in the new clouds. Despite the higher complexity, it may be beneficial to extend the grid refinement scheme directly to the cloud-related microphysics and cover routines in order to simulate a better representation of the cloud condensate and obtain a sizeable improvement in ECHAM-HAM's simulated stratocumulus clouds.

**Table 4.** Global and regional averaged occurrence frequency in the SC-SUND simulation of having a cloud in the ambiguous layer, and of forming a new cloud in the ambiguous layer.

|  | Global | NAM | SAM | AFR | BAR |
|---|---|---|---|---|---|
| Cloud in AL after SC-SUND (%) | 4.1 | 9.9 | 10.8 | 13.3 | 12.2 |
| New cloud formed in AL (%) | 1.8 | 7.1 | 7.9 | 10.2 | 1.9 |

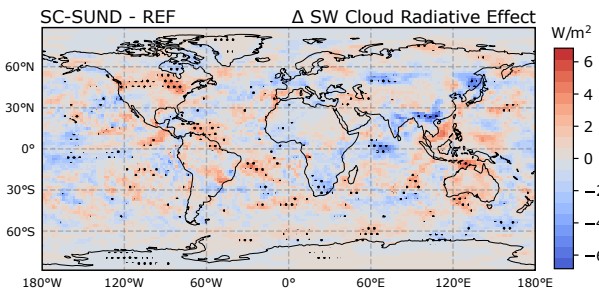

**Figure 10.** SW CRE difference between the REF and SC-SUND simulations. Stippling indicates statistically significant differences at the 95 % significance level; the false discovery rate is controlled following Wilks (2016).

## 4   Summary and conclusions

Two parametrisations for stratocumulus cloud cover based on a vertical grid refinement at the level of the capping inversion were developed and implemented only in the radiation routine of the ECHAM-HAM GCM. SC-VOLUME uses a geometrical and physical argument to augment the cloud's horizontal extent under the inversion; SC-SUND makes use of the improved temperature and water profile at the inversion and re-evaluates the cloud cover.

The inclusion of SC-VOLUME did not lead to significant improvements in the model's cloud cover bias in long-term global climate simulations. Our investigation into the reasons behind this lack of sensitivity (whereas other similar schemes that were also only implemented in the radiation routine have led to improvements in other models, e.g. Boutle and Morcrette, 2010) revealed interesting new insights about ECHAM-HAM's stratocumulus bias that we believe could also be relevant for other models.

Firstly, the simulated stratocumulus clouds are only very rarely occurring in the model layer containing the inversion and appear instead more often in a lower layer. This shows a systematic bias in the model's representation which may be due to poor resolution of the humidity profile not permitting formation of a cloud in the inversion layer. As the correspondence of inversion layer and cloud layer is a necessary condition for the application of SC-VOLUME's cloud squeezing method, this means that the scheme can only be applied in a small fraction of the identified stratocumulus cloud cases, limiting its general effect. Having identified this common stratocumulus-inversion layer mismatch is valuable, as it explains why a geometry-based

method for the representation of stratocumulus clouds such as SC-VOLUME is not widely applicable in ECHAM-HAM and suggests that it would work better with a higher vertical resolution - which would improve stratocumulus cover representation regardless.

Secondly, the SC-MAX experiment showed that even if the cloud cover of all stratocumulus clouds in the model were 100 %, the model's mean cloud cover in stratocumulus regions would still be too low compared to observations. This demonstrates that the model's stratocumulus cover bias is not only due to an underestimation of simulated clouds' horizontal extent, but also to an underestimation of their occurrence frequency and the areas where they appear, i.e. the cloud formation mechanisms are insufficiently parametrised in the first place. Hence, we conclude that a method like SC-VOLUME, which addresses and attempts to correct only cloud amount of pre-existing clouds, is too limited to close the gap.

The SC-SUND scheme aimed to address both of the issues limiting SC-VOLUME. In fact, its application led to the formation of new clouds in the refined below-inversion grid layers and hence to a larger increase of the total cloud cover seen by the radiation routine compared to SC-VOLUME. However this positive effect was not propagated to the simulated cloud cover in a significant form through feedbacks driven by changes in CRE. We showed that the likely reason is that the liquid water content in the newly formed clouds is too low, as it has not been re-calculated with the proper microphysics routine on the new grid. Here as well, the model's original insufficient representation limits the effectiveness of the scheme - despite its numerical advantages, the implementation of a stratocumulus cloud parametrisation limited to the radiation routine is mostly unprofitable.

As the developed grid refinement method itself works well and improves stratocumulus cloud cover within the radiation routine, where it is currently applied, it could be valuable in the future to expand its use to other parts of the model as well. In particular, as a further step the grid refinement could be applied in the cloud microphysics and cloud cover routines. There, the refined grid would lead to an improved reconstruction of the water content profile around the inversion and representation of some stratocumulus-related processes, consolidating the improvements in cloud cover. We think that this implementation could be sufficient for model performance improvements, seeing e.g. the radiation-only implementation of Boutle and Morcrette (2010). However, it is also possible that we might then wish to expand the grid refinement scheme to the vertical mixing to further improve the representation. At that point, a full PBL parametrisation such as that presented in Grenier and Bretherton (2001) or Bretherton and Park (2009) may be a neater solution.

The choice of a way forward should also be weighed against the more straighforward option of increasing the vertical resolution throughout the model: it is difficult for simple parametrisations to generate a better representation when the underlying state is flawed due to poor resolution. This would be 'safer' as it would not involve approximations made when parametrising physical processes, and would also be beneficial for phenomena other than stratocumulus clouds, such as convection. The commonly cited disadvantage of this approach is the computational cost, which is proportional to the number of grid-layers. Also, having a variable interface level matching the inversion and hence allowing for more variable PBL heights has the potential to better represent real situations. It is advantageous for the physics both in terms of cloud location and vertical mixing across the PBL, compared to just having more, but fixed, levels (e.g. Suarez et al., 1983).

Finally, and perhaps most simply, the Sundqvist cloud cover scheme used in ECHAM-HAM is simply not suited for layers representing different air masses with distinct properties, such as those around the inversion in a stratocumulus-capped marine

PBL. It tends to underestimate the cloud cover that would be expected from the moist part, since it assumes the gridbox mean RH is representative of uniform layer conditions. Recently, Weverberg et al. (2021b) developed a cloud cover parametrisation ideally suited to these situations: the cloud fraction and water content are derived from a bimodal probability distribution function representing the sub-grid saturation variations, with a dry and a moist mode. The scheme improves several cloud properties in regional simulations compared to schemes assuming a unimodal PDF, such as implicitly assumed in the Sundqvist

scheme (Weverberg et al., 2021a). For improving stratocumulus cloud cover in ECHAM-HAM, reconsidering the cloud fraction scheme itself and updating it to a more physical one could also be a productive step forward.

## Appendix A: Illustrative radiative transfer calculations with *invgrid*

To illustrate the radiative effect of cloud squeezing with *invgrid* (SC-SUND), we performed a simple radiative flux calculation. We consider only the shortwave (SW) radiative flux for simplicity, as it is the dominant factor. In ECHAM-HAM, the radiative

flux through a column grid-box observed at the top of the atmosphere (TOA) is:

$$F_{allsky} = (1 - b) \, F_{clear} + b \, F_{cloud} \tag{A1}$$

where $b$ denotes the layer cloud cover. The SW cloud radiative effect (CRE) is defined as:

$$CRE_{SW} = F_{allsky} - F_{clear} \tag{A2}$$

We calculate the clear-sky and cloudy shortwave fluxes using radiative transfer equations from Corti and Peter (2009):

$$F_{clear}^{SW} \approx I_0 \left( 1 - r - tt^{'} \alpha \right) \tag{A3}$$

$$F_{cloud}^{SW} \approx I_0 \left( 1 - r - tt^{'} \alpha - (1 - \alpha) \, tt^{'} \frac{R_c - \alpha R_c^{'}}{1 - \alpha R_c^{'}} \right) \tag{A4}$$

where $I_0$ is the incoming solar flux, $r$ is the atmospheric reflectivity, $tt^{'}$ is the product of downward and upward atmospheric transmittance, $\alpha$ is the surface albedo, and $R_c$ and $R_c^{'}$ are the cloud reflectances for incoming and outgoing radiation respectively, calculated as

$$R_c \approx \frac{\tau / \zeta}{\gamma + \tau / \zeta} \tag{A5}$$

$$R_c^{'} \approx \frac{2\tau}{\gamma + 2\tau} \tag{A6}$$

where $\tau$ is the cloud optical depth, $\gamma = 1/(1 - g)$ and $g$ is the asymmetry factor, and $\zeta$ is the cosine of the solar zenith angle (SZA).

    In our calculations, the values used for the parameters are $r = 0.15$, $tt^{'} = 0.73$ and $\gamma = 0.77$ (Corti and Peter, 2009), $\alpha = 0.05$

for the ocean, SZA $= 45°$ and $I_0 = 1360$ W m$^{-2}$ (solar constant).

    We use as an example $\tau = 12$ and $b = 0.6$. If the layer thickness $Z$ is reduced by one third with invgrid ($Z^{ig}/Z = 2/3$), on the refined grid (denoted with superscript $ig$) we obtain $b^{ig} = b \cdot Z/Z^{ig} = 0.9$ and $\tau^{ig} = \tau \cdot b/b^{ig} = 8$. The resulting all-sky

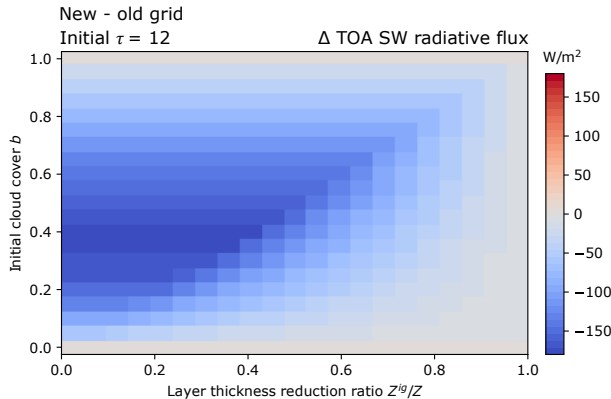

**Figure A1.** Change in SW radiative flux due to application of SC-VOLUME cloud squeezing for a cloud with an initial $\tau = 12$, varying initial cloud cover values as on the y-axis and varying thickness reduction ratios as on the x-axis.

fluxes and SW CRE are

$$F_{allsky} \approx 724 \text{ W m}^{-2} \tag{A7}$$

$$F_{allsky}^{ig} \approx 613 \text{ W m}^{-2} \tag{A8}$$

$$CRE_{SW} \approx -382 \text{ W m}^{-2} \tag{A9}$$

$$CRE_{SW}^{ig} \approx -493 \text{ W m}^{-2} \tag{A10}$$

i.e. the effect of cloud squeezing is a reduction in the net shortwave radiative flux at the TOA.

Figure A1 shows the difference between new and old grid shortwave radiative flux, as a function of initial layer cloud cover and layer thickness reduction ratio. Applying the SC-VOLUME scheme can be seen to always have a negative (or at most zero) SW radiative effect. From the results presented in Table 3, in SC-VOLUME on average the affected layers in stratocumulus regions have initial values of $b = 0.75$ and $Z^{ig}/Z = 0.95$. In these conditions, with an initial value of $\tau = 12$ as in Fig. A1, the SW radiative effect would be relatively small, with around $-20 \text{ W m}^{-2}$.

## Appendix B: Modified EPIC SCM experiments

We ran the EPIC SCM experiment with modified forcing conditions and perturbed initial fields to further test the performance of the inversion reconstruction method. In our main EPIC SCM simulations, the measured values of the vertical temperature and humidity profiles were used in the model at every timestep ('fully forced'). This allowed us to focus on analysing the performance of the inversion reconstruction scheme while being minimally biased by the SCM's ability to accurately reproduce a situation from a forcing. With this setup, a perturbation of the initial conditions (as in Hack and Pedretti (2000)) would dissipate in a few timesteps. Therefore, for the perturbation experiments, we weakly nudged the SCM instead, with a relaxation timescale $\tau_x$ of 5 hours for temperature and humidity (see Eq. 25 in Lohmann et al. (1999)). The initial temperature and absolute

**Table B1.** Results from the modified EPIC SCM experiments: percentage of the time the inversion is reconstructed, and mean and standard deviation of the mismatch (absolute difference) between the reconstructed inversion height $z_{inv}$ and the measured cloud top (linearly interpolated to all model timesteps). Results are presented for the fully forced simulation, nudged simulations with $\tau_x$ of 4, 5 and 6 hours, and for the lower and upper quartiles of the 50 perturbed simulations of the $\tau_x = 5$ h experiment.

| | Fully forced | $\tau_x = 4$h | $\tau_x = 5$h | $\tau_x = 6$h | Pert. Q1 | Pert. Q3 |
|---|---|---|---|---|---|---|
| Inversion found (%) | 78.8 | 78.8 | 78.8 | 78.8 | 78.8 | 78.8 |
| Mean $z_{inv}$ mismatch (m) | 83.7 | 126.8 | 134.9 | 133.8 | 133.8 | 143.3 |
| St. dev. of $z_{inv}$ mismatch (m) | 64.2 | 101.7 | 119.5 | 122.0 | 117.5 | 125.0 |

humidity fields were perturbed following Hack and Pedretti (2000), i.e. for the temperature an normal additive perturbation with standard deviation of 0.5 K and absolutely bounded by 0.9 K, and for the absolute humidity a multiplicative perturbation such that the standard deviation is 0.5 g kg$^{-1}$ in the boundary layer and absolutely bounded by 6 % were used. The experiment with $\tau_x = 5$ h was run 50 times with different perturbed initial conditions. With this nudging setup, perturbations in the initial conditions can lead to differences in the reconstructed inversion height throughout the duration of the simulation.

The results are shown in Fig. B1. Compared to the previous fully forced simulation, the results appear less accurate at times, with more frequent sudden "jumps", particularly around the third day. However, the reconstructed inversion height is well in line with the observed cloud top during the first two and last two days. When the initial conditions are perturbed, the results for the inversion height deviate from the unperturbed simulation mostly during the first day, with only a few also deviating around days 4 and 5. Overall, despite the evolution being more weakly forced, the reconstructed inversion height remains mostly consistent. This suggests that the inversion reconstruction method is robust.

In Table B1, some statistical characteristics of the simulations are given. Note that the percentage of the time the inversion is successfully reconstructed remains constant across the simulations because the stability criterion used depends also on the large scale subsidence, which is unaffected by $\tau_x$ or by the perturbations in temperature and humidity, which in the case of the EPIC SCM experiment proves to be the limiting factor.

*Code and data availability.* The ECHAM-HAMMOZ model is made freely available to the scientific community under the HAMMOZ Software License Agreement, which defines the conditions under which the model can be used. The specific version of the code used for this study is archived in the ECHAM-HAMMOZ SVN repository at `/root/echam6-hammoz/tags/papers/2021/Pelucchi_et_al_GMDD`. More information can be found at the HAMMOZ website (https://redmine.hammoz.ethz.ch/projects/hammoz, last access: 2 May 2021). The data used to produce the figures in this manuscript can be found at https://doi.org/10.5281/zenodo.4268194; scripts can be found at https://doi.org/10.5281/zenodo.4268168. The Calipso-GOCCP product can be obtained from http://climserv.ipsl.polytechnique.fr/cfmip-obs/. The EPIC campaign data can be obtained from https://atmos.washington.edu/~breth/EPIC/EPIC2001_Sc_ID/sc_integ_data_fr.htm (last access: 2 May 2021) (Bretherton, 2005). The observational climatology of cloud occurrence frequency can be obtained from https://atmos.uw.edu/CloudMap/ (Hahn and Warren, 2007; Eastman et al., 2014).

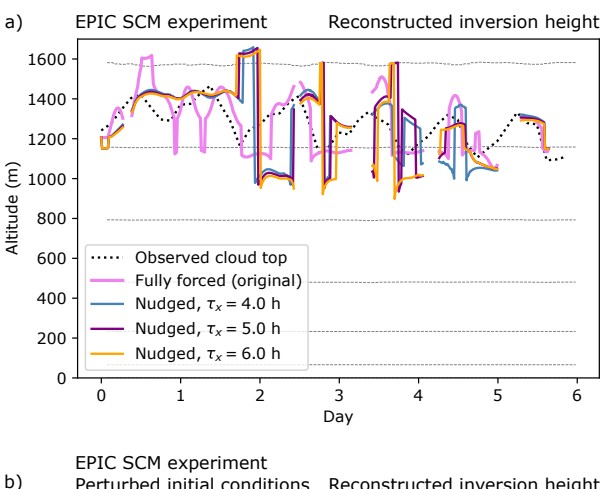

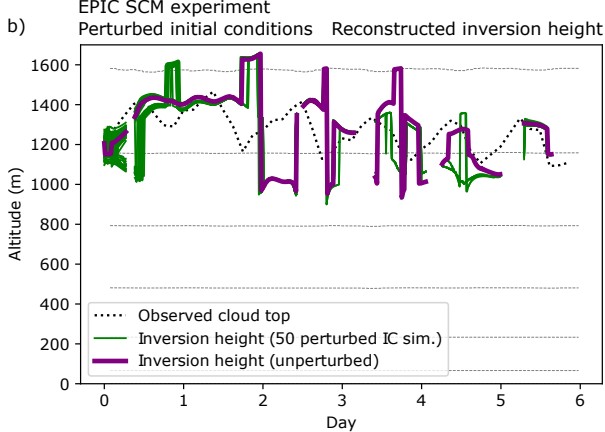

**Figure B1.** Reconstructed inversion heights in modified EPIC SCM experiments, (a) with different nudging relaxation timescales $\tau_x$; (b) with $\tau_x = 5$h and perturbed initial conditions.

*Author contributions.* DN conceived the idea for the study. PP designed the experiments with help from DN and UL. PP implemented the vertical grid refinement schemes and conducted the simulations. PP analysed the results with contributions from DN and UL. PP wrote the paper with comments from DN and UL.

*Competing interests.* The authors declare that they have no conflict of interest.

*Acknowledgements.* The ECHAM-HAMMOZ model is developed by a consortium composed of ETH Zurich, Max Planck Institut für Meteorologie, Forschungszentrum Jülich, University of Oxford, the Finnish Meteorological Institute and the Leibniz Institute for Tropospheric Research, and managed by the Center for Climate Systems Modeling (C2SM) at ETH Zurich. We would like to thank Colombe Siegenthaler-Le Drian for her previous work in the model and for the helpful discussions.

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
