# Peer review of "Vertical grid refinement for stratocumulus clouds in the radiation scheme of the global climate model ECHAM6.3-HAM2.3-P3"

_Geoscientific Model Development, 2020_

## Author Comment (AC1)

**Author Response to Reviews of**

**Vertical grid refinement for stratocumulus clouds in the radiation scheme of a global climate model**

Paolo Pelucchi, David Neubauer, and Ulrike Lohmann

*Geoscientific Model Development,* `doi:10.5194/gmd-2020-384`
* * *
RC: *Reviewer Comment*,     AR: *Author Response*,     ☐ Manuscript text

**1. Reviewer Comment #1**

RC:  *In this paper, two related approaches to improve biases in stratocumulus cloud cover in ECHAM-HAM are presented. They both rely on a vertical grid refinement step. This allows the determination of the inversion level and then a remapping step, where the two methods differ. The updated vertical profiles are used in the radiation calculation. This is an interesting study. The results are basically negative, but are presented well. Given the nature of the results, I do not think it is incumbent upon the authors nor reviewers to try to "fix" these schemes or to subject the schemes to much additional scrutiny in this paper. The text identifies some of the difficulties with the approach. In my estimation, these boil down to (1) adjusting the cloud cover just for the radiation calculation does not seem to be enough to push the model toward a more realistic climate (in terms of stratocumulus cover) and (2) there are probably too many concessions in only applying the vertical refinement to the gridscale clouds to make it worth the effort for minimal improvements to the climate. I think only minor revisions are need for this paper.*

AR:  *Thank you for your feedback. Please find our responses to your general points and specific comments below.*

**General points**

RC:  *There is an opportunity here to make the work more broadly applicable by (1) making some comparison to other global models that might use similar approaches and (2) possibly making more assertive statements about what are likely to be productive paths toward improved stratocumulus representations in global models. On these two general points, I'll just add a couple thoughts.*

**Point (1)**

RC:  *I noted in my specific comments below a connection two two other models. First is the NCAR CAM5 that used the UW moist turbulence and shallow convection schemes, which I think are based on Grenier and Bretherton. It would be interesting to know whether the results here could be related to results with that model (where the ambiguous layer must be used for determining turbulent mixing and possibly cloud cover). The other is the UCLA model that used a mixed layer model for their boundary layer scheme; that model isn't really relevant any more, but there are some shared assumptions with the scheme used here, so I wondered if there was any merit in making that connection?*

AR:  *Thank you for indicating these two models to us. The approaches used in these studies to model the stratocumulus-capped boundary layer do have a connection to our method, and the results obtained better inform the discussion around our own results. We have added a few sentences presenting these studies in*

*our Introduction section; we further reference them in the context of our discussion of recommendations for future work.*

**Point (2)**

RC: ***On the second point, I think the conclusions here could be expanded a little bit. In particular, I wonder whether any recommendations could be made beyond the possibility of also applying the refined grid to the microphysics. Below I note some skepticism about that path, as it would seem to just lead to wanting to apply the refined grid to the rest of the physics too. One could also ask whether there is value in trying to better match the physics and dynamics by better including the inversion reconstruction in the vertical advection? Finally, and maybe most simply, how much benefit would there be to just increasing the vertical resolution in the boundary layer versus trying to reconstruct the smaller-scale structure?***

AR: *We agree this discussion deserves more consideration. We have expanded our Conclusion section with a discussion of potential advantages and disadvantages of applying the* invgrid *grid refinement scheme to other routines, of having a variable model level corresponding to the boundary layer top, and of using fixed grid levels but at higher resolution in the boundary layer.*

**Specific comments**

**1.1. Around line 105**

RC: ***The "invgrid" with the squished clouds seems like it would be problematic from the outset. Since the optical depth of the clouds is directly related to the LWP, when the cloud volume is conserved but made geometrically thinner but broader, the cloud fraction goes up, but the LWP would be reduced, correct? Seems like that could end up radiatively warming (via shortwave) the subcloud layer (a small effect) and the surface (potentially large effect in coupled settings, and would work against the goal of increasing cloud cover).***

AR: *The liquid water path (LWP) of the grid-layer is conserved: the layer thickness is reduced, but the liquid water mixing ratio is proportionally increased. The in-cloud LWP, which is what is used for cloudy-sky radiative calculations, is indeed reduced, by the same proportion that the cloud fraction is increased (we have corrected this in the manuscript). This is not problematic, as it is simply a more realistic representation of the situation. Also, the radiative flux calculation is linear in the cloud fraction but non-linear in the LWP (equivalently cloud optical depth). Therefore, there can be a difference in radiative fluxes by applying the* invgrid *scheme.*

*To better illustrate this point, we performed a simple radiative flux calculation. We consider only the shortwave (SW) for simplicity, as it is the dominant factor. In ECHAM-HAM, the radiative flux through a column grid-box observed at the top of the atmosphere (TOA) is:*

$$F_{allsky} = (1 - b) \, F_{clear} + b \, F_{cloud} \tag{1}$$

*where $b$ denotes the layer cloud cover. The SW cloud radiative effect (CRE) is defined as:*

$$CRE_{SW} = F_{allsky} - F_{clear} \tag{2}$$

*We calculate the clear-sky and cloudy shortwave fluxes using radiative transfer equations from Corti and*

*Peter (2009):*

$$F_{clear}^{SW} \approx I_0 \left( 1 - r - tt^{'} \alpha \right) \tag{3}$$

$$F_{cloud}^{SW} \approx I_0 \left( 1 - r - tt^{'} \alpha - (1 - \alpha) \, tt^{'} \frac{R_c - \alpha R_c^{'}}{1 - \alpha R_c^{'}} \right) \tag{4}$$

*where $I_0$ is the incoming solar flux, $r$ is the atmospheric reflectivity, $tt^{'}$ is the product of downward and upward atmospheric transmittance, $\alpha$ is the surface albedo, and $R_c$ and $R_c^{'}$ are the cloud reflectance for incoming and outgoing radiation respectively, calculated as*

$$R_c \approx \frac{\tau/\zeta}{\gamma + \tau/\zeta} \tag{5}$$

$$R_c^{'} \approx \frac{2\tau}{\gamma + 2\tau} \tag{6}$$

*where $\tau$ is the cloud optical depth, $\gamma = 1/(1 - g)$ and $g$ is the asymmetry factor, and $\zeta$ is the cosine of the solar zenith angle (SZA). In our calculations, the values used for the parameters are $r = 0.15$, $tt^{'} = 0.73$ and $\gamma = 0.77$ (Corti and Peter, 2009), $\alpha = 0.05$ for the ocean, SZA $= 45°$ and $I_0 = 1360 \, \mathrm{W\,m^{-2}}$ (solar constant).*

*We use as an example $\tau = 12$ and $b = 0.6$. If the layer thickness $Z$ is reduced by one third with* invgrid *($Z^{ig}/Z = 2/3$), on the refined grid (denoted with superscript ig) we obtain $b^{ig} = b \cdot Z/Z^{ig} = 0.9$ and $\tau^{ig} = \tau \cdot b/b^{ig} = 8$. The resulting all-sky fluxes and SW CRE are*

$$F_{allsky} \approx 724 \, \mathrm{W\,m^{-2}} \tag{7}$$

$$F_{allsky}^{ig} \approx 613 \, \mathrm{W\,m^{-2}} \tag{8}$$

$$CRE_{SW} \approx -382 \, \mathrm{W\,m^{-2}} \tag{9}$$

$$CRE_{SW}^{ig} \approx -494 \, \mathrm{W\,m^{-2}} \tag{10}$$

*i.e. the effect of cloud squeezing is a reduction in the shortwave radiative flux reaching the surface.*

*To give a more general picture, we also investigated the influence of the initial layer cloud cover and of the thickness reduction ratio on the radiative effect of* invgrid. *Figure 1.1 shows the difference between new and old grid SW radiative flux.*

*Applying SC-VOLUME can be seen to always have negative (or at most null) SW radiative effect. From the results presented in Table 3 of the manuscript, on average the affected layers in stratocumulus regions have initial $b = 0.75$ and $Z^{ig}/Z = 0.95$. In these conditions the SW radiative effect is relatively small, around $-20 \, \mathrm{W\,m^{-2}}$.*

**1.2. Around line 130**

RC: ***The column detection method is fine, but is this the complete description? Is there an ocean mask or a latitude limiter applied? Otherwise, it seems like non-subtropical-stratocumulus would be selected often (at high latitudes, for example). Oh, I see that later in the results, the changes outside the subtropics are noted. Was there any attempt to more directly limit the application to stratocumulus?***

AR: *We did not explicitly limit the regions of applicability of the method. The criterion used to select the columns in which to apply the method is based only on the low-tropospheric stability as a measure of the inversion*

[Figure]

Figure 1.1: Change in SW radiative flux due to application of SC-VOLUME cloud squeezing for a cloud with initial $\tau = 12$, initial $b$ as on the y-axis and a thickness reduction ratio as on the x-axis.

*strength. We chose to keep the criterion general and purely physics-based because, as the model struggles to represent sharp inversions in general, we believe that applying the method can be beneficial in any column with this problem, and not only for stratocumulus.*

**1.3. Line 160**

**RC:** *A small notational thing, I always reserve q for specific humidity while r is used for mass mixing ratio. It's a convention that is known, but not always followed. I just want to confirm that it is mass mixing ratio that is being used (mass_water / mass_dry_air) and not specific humidity (mass_water / (mass_dry_air + Sum_i(mass_water_i))).*

AR: *Thank you for pointing this out. We are indeed using mass mixing ratio, and accordingly changed the notation to r in the manuscript.*

**1.4. Line 200**

**RC:** *In climatologies stratocumulus will be thicker, but instantaneously, wouldn't we expect to often find much thinner layers?*

AR: *The "climatology" we were referring to from Wood (2012) is actually compilation of instantaneous thicknesses, rather than climatological thicknesses - we have corrected this in the manuscript. We introduced the $50\,\mathrm{m}$ threshold to avoid situations in which the reconstructed inversion would be found and/or get stuck on a half-level, and so the cloud would be almost infinitesimally thin, in order to avoid numerical problems and non-physical values.*

**1.5. Line 249**

**RC:** *I don't think that "heat content H" is an appropriate thermodynamic description for Equation 13; isn't this more correctly called the enthalpy?*

AR: *We agree that 'heat content' is not the appropriate term. By definition 'enthalpy' would also include the*

*pressure-volume product, not present in our equation. We have hence changed it to 'internal energy', denoted by U.*

**1.6. Sec. 2.2.3 (grid refinement)**

**RC:** *I'm interested to know whether regridding the aerosol fields was considered? I don't remember much about how ECHAM-HAM does aerosol, but I assume that there are a number of species that are advected and interact with the clouds and radiation. In some stratocumulus regimes, for example over the southeast Atlantic during biomass burning over central Africa, there are important aerosol direct (and possibly indirect) effects that influence the cloud/boundary layer structure. This seems like it could be problematic for this approach, since the aerosol will interact with clouds separate from the radiation, so it would not be obvious how to regrid the aerosol. If the aerosol is left homogeneously distributed in the grid cell, it could alter the radiative forcing in the column.*

AR: *Aerosols are not regridded. The effect of aerosols on the clouds and boundary layer are still modelled normally, e.g. in the microphysics routine, where no grid refinement takes place. In the radiation routine, as they are left untouched in the affected layers, the aerosols' radiative forcing may indeed be slightly altered. However, we believe that the aerosol effects are of secondary importance to the effects due to change in cloud cover and cloud condensate. As we found these to be small, the aerosol radiative change is most likely negligible. Of course, for consistency the regridding of aerosols should be considered. We have added references to this point in Section 2.2.3 and in the outlook.*

**1.7. Sec. 2.2.3 (grid refinement)**

**RC:** *The other question I had was whether rain or snow are radiatively active in the model. If so, it seems like they would need to be regridded as well – for example, to avoid the situation where drizzle is falling into the stratocumulus that made it (!).*

AR: *Precipitation is not radiatively active in the model, so luckily, such situations do not arise.*

**1.8.**

**RC:** *Another question about the scheme itself is what it looks like for stratocumulus that are multiple grid levels deep? I assume this occurs frequently (it does in other models with similar resolution). It would seem like this would impose a structure that would be more like cumulus rising into stratocumulus in some circumstances. Or maybe I missed a detail, and there is some adjustment to the lower cloud layer, too? Later in the paper this is shown a little bit (Figure 5), and is kind of addressed in the discussion of the difference between the VOLUME and SUND schemes, but not completely. In the AMIP runs, I would expect multi-level clouds to occur frequently, and I'm still not sure if anything is done for the lower cloud layer in the case when the inversion level pushes the cloud top down into a level (rather than popping it up to the next level as in Fig 5b and c).*

AR: *The scheme adjusts only the location of the cloud top by pushing the half-level atop of the uppermost cloud layer downwards, and therefore the lower layers of a multi-layer cloud are not affected. Given the restrictions for ambiguous layer choice we implemented (Sec. 2.2.1), it would not be possible to have a case where the inversion is reconstructed in a lower cloud layer leaving an unaffected cloud layer above. To clarify this point, we have added a sentence to explicitly address the treatment of multi-level clouds in Sec. 2.2.3.*
*As a related but separate issue, we considered adjusting the cloud base and attempted doing so by shifting the half-level under the lowermost cloud layer to the lifting condensation level, however we abandoned this idea*

*as our reconstruction of the LCL was not successful (Sec. 3.1).*

**1.9.   L330 / Fig 3**

**RC:**   *The failure of the LCL diagnostic is interesting (although probably secondary to the main topic). It would informative to include in Figure 3 and indication of where "cloud base" is in the actual model. That is, mark the bottom of the level with nonnegligible liquid water. This seems to be indicated in Fig 5, so maybe it isn't worth adding to Figure 3.*

**AR:**   *We have added an indication in the manuscript to point to the model cloud base and top shown in Figure 5.*

**1.10.**

**RC:**   *Also, a 6-day SCM run isn't very convincing in terms of the success of the inversion reconstruction. Were other cases like DYCOMS-II or ASTEX also investigated? At L334, the comparison with previous results is noted. It is hard to have much confidence in this improvement based on what is shown. Another option would be to re-run the EPIC case a bunch of times with perturbed initial conditions (as in Hack & Pedretti 2000) to get a better sense of the statistical properties of the inversion reconstruction.*

**AR:**   *We did the initial testing of the scheme in SCM with an idealised situation (no diurnal cycle) from the CGILS project (Zhang et al., 2013), but this is not shown. We did not investigate other cases due to time constraints and lack of appropriate forcing files adapted to the version of ECHAM-HAM used.*

*Re-running EPIC with perturbed initial condition is an interesting idea, thank you for the suggestion. In our EPIC SCM simulations, the measured values of the vertical temperature and humidity profiles were used in the model at every timestep ('fully forced'). This allowed us to focus on analysing the performance of the inversion reconstruction scheme while being minimally biased by the SCM's ability to accurately reproduce a situation from a forcing. With this setup, a perturbation of the initial conditions (as in Hack and Pedretti (2000)) would dissipate in a few timesteps. Therefore, for the perturbation experiments, we weakly nudged the SCM instead, with a relaxation timescale $\tau_x$ of 5 hours for temperature and humidity (see Eq. 25 in Lohmann et al. (1999)). The initial temperature and absolute humidity fields were perturbed following Hack and Pedretti (2000), i.e. for the temperature an normal additive perturbation with standard deviation of $0.5\,\mathrm{K}$ and absolutely bounded by $0.9\,\mathrm{K}$, and for the absolute humidity a multiplicative perturbation such that the standard deviation is $0.5\,\mathrm{g\,kg^{-1}}$ in the boundary layer and absolutely bounded by $6\,\%$. The experiment was run 50 times with different perturbed initial conditions. With this weak nudging setup, perturbations in the initial conditions can lead to differences in the reconstructed inversion height throughout the duration of the simulation. The new results are shown in Appendix A, Fig. A.1 and Table 1.*
*Compared to the previous fully forced simulation, the results appear less accurate at times, with more frequent sudden "jumps", particularly around the third day. However, the reconstructed inversion height is well in line with the observed cloud top during the first two and last two days. When the initial conditions are perturbed, the results for the inversion height deviate from the unperturbed simulation mostly during the first day, with only a few also deviating around days 4 and 5. Overall, despite the evolution being much more weakly forced, the reconstructed inversion height remains mostly consistent. This suggests that the inversion reconstruction method is robust.*

**1.11.**

**RC:**   *I think Figure 4 should also include the radiosondes from EPIC that are mentioned previously. Looks like the data is available here:* `https://atmos.washington.edu/~breth/EPIC/EPIC2001_Sc_`

*ID/sc_integ_data_fr.htm*

AR: *Thank you for the suggestion. We have added the measured vertical profiles to Figure 4 in the manuscript - see Figure 1.2. We also show the same timestep in the nudged simulation mentioned in the previous point, to illustrate what the effect of the reconstruction might be in a more weakly forced situation where the profile was not already set to the measured one. The chosen timestep turned out not to be ideal to demonstrate the effect, as the real inversion already fell on a model level. However, it shows (especially in the nudged simulation) how even in such situations the grid refinement scheme can improve the shape of the profile.*

**1.12. Around L465**

RC: **This is a key conclusion of the paper, I think. If we think of this vertical regridding scheme as an attempt at some kind of "dynamic bias correction" to cloud cover, it doesn't really work. The initial cloud formation mechanisms are flawed, so a scheme that would just try to boost the cloud cover in the radiation is extremely limited in utility.**

AR: *Thank you, we agree. We have further highlighted this point - about ECHAM-HAM's lack of stratocumulus clouds in the first place - in the conclusion. We would also like to note however that improving the cloud cover in the radiation routine has worked in other models to improve other fields, e.g. Boutle and Morcrette (2010).*

**1.13. L502-4**

RC: **I agree that one might expect better performance by also applying the regridding to the the microphysics. That would be like an improved version of the SCSUND scheme that would deal with phase partitioning and drop numbers better (and aerosol?). I would suggest that approach would also come up short, and that the argument then would be that the turbulent mixing isn't represented correctly because it doesn't know the correct "mixing height" because it is acting (probably, depends on the scheme) on full model layers. So if the microphysics and radiation were adjusted, the recommendation might be to extend the adjustment to the turbulence, too. At that point, the Grenier and Bretherton (and then Bretherton and Park / Park and Bretherton) schemes would seem like an attractive solution, harmonizing the shallow convection, gridscale cloud physics, and turbulence; the microphysics and radiation then get to come along for the ride, but would depend a bit on the implementation. I don't think that in NCAR-CAM5 (which uses the Bretherton/Parks schemes) the radiation has any information about the inversion height.**

AR: *This is a valid point. However, we would not exclude a priori that an implementation in the radiation, cloud cover and microphysics routine could work, but it is worth mentioning that even then performance could be unsatisfactory because the information about the inversion is not being used in the vertical mixing scheme. In that case, a full PBL parametrisation scheme such as Grenier and Bretherton (2001) may be preferable. We have added a discussion of these points in our manuscript, also in the context of our response to general point (2) above.*

*In the past an attempt was made to implement a scheme similar to* invgrid *(but where two new variable levels were added) fully interactively in ECHAM-HAM (see Siegenthaler-Le Drian (2010)) but the results were unsuccessful due to unspecified numerical problems. A study by Boutle and Morcrette (2010) had obtained an improvement in global climate fields while implementing a simple stratocumulus cloud cover parametrisation only in the radiation scheme. This had encouraged us to investigate whether similar results could be obtained in ECHAM-HAM, however in our case we found the issue of the cloud layer mismatch, discussed in Sec. 3.3.1, particularly problematic for our approach.*

Figure 1.2: Vertical profiles at the end of the fifth day (timestep 471) of the EPIC SCM experiments. Top: experiment with fully forced profiles; bottom: nudged experiment with 5-hour relaxation timescale.

**RC:** *Another model that I thought about while reading this paper was the old UCLA GCM. The relevant idea there was that they used a well-mixed layer assumption to determine their lowest model level's height, which was synonymous with the "boundary layer". They did a relatively good job with stratocumulus because they had a level interface that was naturally at the inversion. (Now, where the mixed layer assumption didn't work well raised other important errors, but for stratocumulus it worked pretty well.) See Suarez et al. (1983) and Randall and Suarez (1984).*

**AR:** *Thank you for the references. We have added a few sentences referencing and discussing this model to our manuscript, as stated in our response to general point (1) above.*

**2. Reviewer Comment #2**

**RC:** *This paper describes the application of grid-refinement techniques to improve the cloud cover under inversions as seen by the radiation scheme in the ECHAM-HAM model. Ultimately, the attempts are unsuccessful in improving the mean model climate, which is somewhat sad as the paper is very well written. It leaves me quite uncertain what to suggest. On one hand, I'm supportive of publishing a study like this, as it is useful to the community to know what has been done, and that (in this case) it doesn't really work. On the other, I'm unsure that the paper contains enough new material to be published. In particular:*

- *The method of grid refinement is not new, it is simply an application of an already published study (Grenier & Bretherton 2001).*

- *The idea of giving the radiation scheme the spatial area of cloud seen rather than the volume fraction is not new, but has been discussed by several previous studies (most recently Boutle & Morcrette 2010).*

- *Applying grid-refinement techniques to improve cloud cover has been more successfully implemented in other models, and so the application of this to a full GCM is not new. Further to this, the application in other models (e.g. Boutle & Morcrette 2010) has applied the technique to cloud variables throughout the model, rather than just to those seen by radiation. Therefore the previous studies on the topic seem to offer a more complete and consistent solution to the problem, and possibly unsurprisingly, have been more successful in demonstrating model improvements.*

*Therefore I'm struggling to see really what the new results being presented here are.*

**AR:** *Thank you for your feedback. We think that our study presents new ideas and results in the method used for obtaining the new-grid cloud cover in SC-VOLUME and especially in the insights about ECHAM-HAM's representation of stratocumulus clouds obtained from our analysis of **why** the proposed approaches had such little effect. As you mentioned, we used the inversion reconstruction method by Grenier and Bretherton (2001) to find the location of the inversion and model the sub-grid thermodynamic profiles. However, the 'cloud squeezing' method applied to recalculate the new-grid cloud cover in SC-VOLUME, based on the simple and physically-motivated idea of using the inversion as the cloud top and conserving the cloud volume to obtain the real simulated horizontal cloud fraction, is a novel idea to our knowledge. In Boutle and Morcrette (2010), for example, the new horizontal cloud fraction used is the maximum value obtained from estimating the volume fraction on the three sub-levels onto which the inversion is sharpened by extrapolation. Additionally, in Boutle and Morcrette (2010), the new cloud cover was used only in the radiation routine ("The interpolation/*

extrapolation is only used to generate a value of $C_a$ to pass to the radiation scheme and for use in diagnostic outputs; it is not directly communicated to other parts of the model.", *Boutle and Morcrette (2010), Sec. 2.2), but improvements in other climatological fields were observed thanks to feedbacks. We had initially hoped that an implementation of the scheme limited to the radiation routine would be sufficient or beneficial in our model as well. Even though this turned out not to be the case, we believe that our subsequent investigation into the reasons behind the scheme's lack of success revealed interesting new insights about ECHAM-HAM's stratocumulus bias that could also be relevant for other models; most notably the cloud-inversion layer mismatch and the insufficient occurrence of stratocumulus clouds in the first place.*

**2.1.  Suggestion 1.**

**RC:**  ***The best suggestion I can offer is to try the experiment applying SC-SUND to all cloud, not just that seen by the radiation. This would be consistent with how previous studies have applied similar techniques and demonstrated improvement. It would seem that you've done all the hard work in coding up the new scheme, and therefore linking it in to the main cloud water/fraction variables is a trivial extra step. This would (hopefully) not only allow you to show a model improvement that ECHAM-HAM developers/users would be interested in, but also allow discussion of why only applying the scheme to the radiation does not work.***

 AR:  *We think that applying the cloud cover calculated by the SC-SUND scheme for the refined grid elsewhere in the model would be problematic, as it would be inconsistent with the representations in the respective routines. For example, the microphysics routine is written with an interpretation of the cloud cover as a volume fraction, and hence using the SC-SUND cloud cover without also accordingly reducing the vertical resolution would result in an increase in cloud volume inconsistent with the microphysical calculations previously carried out. On the other hand, applying the grid refinement to other routines is a major undertaking and would require rewriting large parts of them. This is outside of the scope of this study, which was an attempt at investigating whether improving only the radiatively active cloud cover could improve it in the model in general.*

**2.2.  Suggestion 2.**

**RC:**  ***I feel discussion of this point is somewhat lacking in the current paper. The expectation is clearly that this is the most important term in the cloud budget, and therefore should be sufficient - so why isn't it? It looks from Fig 6 that the increase in cloud from SC-SUND (e) is almost comparable to the bias in main model cloud (b). So is having improved radiative fluxes in these regions (I assume they are improved - this is something else that could be shown and discussed in the paper) not feeding back onto the inversion structure in a way that allows the cloud to form properly there? Or is the model vertical grid so coarse and inadequate that there is no hope of ever forming cloud correctly there? Both of these would clearly motivate diagnosing the full model cloud quantities using SC-SUND, as this will compensate for the poor vertical resolution, but also allow further improvements to the radiative fluxes and inversion structure, feeding back onto the cloud properties.***

 AR:  *We briefly discussed the reasons for the low radiative effect of SC-SUND in Sec. 3.3.3, but we agree that this point is deserving of more examination. Our proposed explanation is that the clouds newly formed with SC-SUND have too-low water condensate, because their water content is derived from the original grid where no cloud was present, so that they cannot produce a significant radiative difference.*

*To further motivate this explanation, we looked at the difference in SW cloud radiative effect (CRE) between the REF and SC-SUND simulations, which can give us some information about whether the new clouds formed in SC-SUND are radiatively different from the 'original' ones. As the CRE is the difference between all-sky and clear sky radiative flux, its magnitude can change based on changes in cloud cover, cloud occurrence*

*frequency or cloud optical thickness. If the mean cloud optical thickness remains the same, an increase in mean cloud cover in stratocumulus (Sc) regions as observed by the radiation routine in SC-SUND (both cloud cover and cloud frequency increased) should result in a more negative CRE in those regions. The following plot (Figure 2.1) shows the difference in mean SW CRE between the REF and SC-SUND simulations. Stippling shows regions where the difference is statistically significant at the 95% significance level.*

[Figure]

Figure 2.1: SW CRE difference between the REF and SC-SUND simulations.

*In Sc regions, where an important increase in mean cloud cover took place, the difference is still not significant. This can only be explained if the mean optical thickness of clouds in SC-SUND was much lower, compensating in the radiative effect the increase in cloud cover and occurrence. This provides an explanation as to why application of SC-SUND only in the radiation routine did not have the desired effect despite the large increase in cloud cover and occurrence - as suggested, the newly formed clouds are devoid of significant cloud condensate. If the new clouds were comparable to the usual clouds in terms of water condensate, a strong radiative change would be observed, leading to favourable feedbacks, like in SC-MAX. This motivates extending the* invgrid *grid refinement to the microphysics routine, which would then simulate a better representation of the cloud condensate to be passed the radiation routine.*

**2.3. Suggestion 3.**

**RC:** *My other suggestion would be to link the discussion to recent literature a bit more. Sundqvist-type cloud schemes that use a critical relative humidity are somewhat arcane and will always struggle around inversions due to the mixing of boundary-layer and free tropospheric air masses in a way that cannot be represented by a simple monomodal PDF and critical relative humidity. A (very) recent set of papers (van Weverberg et al. 2021a,b) has discussed this in detail, demonstrating that really the cloud properties here need to be considered as bimodal, and representing them otherwise probably places fundamental limits on how good the cloud can ever be near an inversion.*

AR: *Thank you very much for the recommendation. The idea proposed in Weverberg et al. (2021b) and Weverberg et al. (2021a) could be a viable alternative approach to improve Sc cloud cover in layers that represent a mixture of dry and moist air, and where hence the Sundqvist approach underestimates it, so it is definitely worth discussing. We have added a reference to these studies and discussion in relation to ours to our manuscript.*

**3.  Chief Editor Comment**

RC:  *Dear authors,*
*in my role as Executive editor of GMD, I would like to bring to your attention our Editorial version 1.2:*
`https://www.geosci-model-dev.net/12/2215/2019/`
*This highlights some requirements of papers published in GMD, which is also available on the GMD website in the 'Manuscript Types' section:*
`http://www.geoscientific-model-development.net/submission/manuscript_types.html`
*In particular, please note that for your paper, the following requirements have not been met in the Discussions paper:*

AR:  *Thank you for bringing these requirements to our attention. We apologise for not meeting them in the Discussion paper.*

**3.1.**

RC:  *The main paper must give the model name and version number (or other unique identifier) in the title.*

AR:  *We have changed the title of the manuscript to*

> *Vertical grid refinement for stratocumulus clouds in the radiation scheme of the global climate model ECHAM6.3-HAM2.3-P3*

**3.2.**

RC:  *"Code must be published on a persistent public archive with a unique identifier for the exact model version described in the paper or uploaded to the supplement, unless this is impossible for reasons beyond the control of authors. All papers must include a section, at the end of the paper, entitled "Code availability". Here, either instructions for obtaining the code, or the reasons why the code is not available should be clearly stated. It is preferred for the code to be uploaded as a supplement or to be made available at a data repository with an associated DOI (digital object identifier) for the exact model version described in the paper. Alternatively, for established models, there may be an existing means of accessing the code through a particular system. In this case, there must exist a means of permanently accessing the precise model version described in the paper. In some cases, authors may prefer to put models on their own website, or to act as a point of contact for obtaining the code. Given the impermanence of websites and email addresses, this is not encouraged, and authors should consider improving the availability with a more permanent arrangement. Making code available through personal websites or via email contact to the authors is not sufficient. After the paper is accepted the model archive should be updated to include a link to the GMD paper."*

AR:  *We have added a link to the SVN tag corresponding to the specific model version used to the* Code and data availability *section:*

> *The ECHAM-HAMMOZ model is made freely available to the scientific community under the HAMMOZ Software License Agreement, which defines the conditions under which the model can be used. The specific version of the code used for this study is archived in the ECHAM-HAMMOZ SVN repository at `/root/echam6-hammoz/tags/papers/2021/Pelucchi_et_al_GMDD`. More information can be found at the HAMMOZ website (https://redmine.hammoz.ethz.ch/projects/hammoz, last access: 21 September 2020).*

*For the final paper a new SVN tag will be added and this link will be updated. The code can be accessed if the HAMMOZ License, which is free for research, is obtained.*

**RC:** ***Please note, that even though the code is not freely available, the exact code version used for the publication needs to be archived. There please provide an identifyer or other means how the exact code version can be accessed.***
***Please add the name and version number of the model used (ECHAM-HAMMOZ) and the version number to the title of your manuscript.***
***Yours,***
***Astrid Kerkweg***

AR: *We hope that our modifications and additions will be considered in line with the requirements.*

**References**

Boutle, I. A. and C. J. Morcrette (2010). "Parametrization of area cloud fraction". In: *Atmospheric Science Letters* 11.4, pp. 283–289. DOI: 10.1002/asl.293. eprint: https://rmets.onlinelibrary.wiley.com/doi/pdf/10.1002/asl.293. URL: https://rmets.onlinelibrary.wiley.com/doi/abs/10.1002/asl.293.

Corti, T. and T. Peter (2009). "A simple model for cloud radiative forcing". In: *Atmospheric Chemistry and Physics* 9.15, pp. 5751–5758. DOI: 10.5194/acp-9-5751-2009. URL: https://acp.copernicus.org/articles/9/5751/2009/.

Grenier, Hervé and Christopher S. Bretherton (2001). "A Moist PBL Parameterization for Large-Scale Models and Its Application to Subtropical Cloud-Topped Marine Boundary Layers". In: *Monthly Weather Review* 129.3, 357–377. DOI: 10.1175/1520-0493(2001)129<0357:amppfl>2.0.co;2.

Hack, James J. and John A. Pedretti (2000). "Assessment of Solution Uncertainties in Single-Column Modeling Frameworks". In: *Journal of Climate* 13.2, pp. 352 –365. DOI: 10.1175/1520-0442(2000)013<0352:AOSUIS>2.0.CO;2. URL: https://journals.ametsoc.org/view/journals/clim/13/2/1520-0442_2000_013_0352_aosuis_2.0.co_2.xml.

Lohmann, Ulrike et al. (1999). "Comparing Different Cloud Schemes of a Single Column Model by Using Mesoscale Forcing and Nudging Technique". In: *Journal of Climate* 12.2, 438–461. DOI: 10.1175/1520-0442(1999)012<0438:cdcsoa>2.0.co;2.

Siegenthaler-Le Drian, Colombe (2010). "Stratocumulus clouds in ECHAM5-HAM". PhD thesis. ETH Zürich.

Weverberg, Kwinten Van, Cyril J. Morcrette, and Ian Boutle (2021a). "A Bimodal Diagnostic Cloud Fraction Parameterization. Part II: Evaluation and Resolution Sensitivity". In: *Monthly Weather Review* 149.3, pp. 859 –878. DOI: 10.1175/MWR-D-20-0230.1. URL: https://journals.ametsoc.org/view/journals/mwre/149/3/MWR-D-20-0230.1.xml.

Weverberg, Kwinten Van et al. (2021b). "A Bimodal Diagnostic Cloud Fraction Parameterization. Part I: Motivating Analysis and Scheme Description". In: *Monthly Weather Review* 149.3, pp. 841 –857. DOI: 10.1175/MWR-D-20-0224.1. URL: https://journals.ametsoc.org/view/journals/mwre/149/3/MWR-D-20-0224.1.xml.

Wood, Robert (2012). "Stratocumulus Clouds". In: *Monthly Weather Review* 140.8, 2373–2423. DOI: 10.1175/mwr-d-11-00121.1.

Zhang, Minghua et al. (2013). "CGILS: Results from the first phase of an international project to understand the physical mechanisms of low cloud feedbacks in single column models". In: *Journal of Advances in Modeling Earth Systems* 5.4, 826–842. DOI: 10.1002/2013ms000246.

**A.  Appendix: Modified EPIC SCM experiment**

Table 1: Results from the modified EPIC SCM experiments: percentage of the time the inversion is reconstructed, and mean and standard deviation of the mismatch (absolute difference) between the reconstructed inversion height $z_{inv}$ and the measured cloud top (linearly interpolated to all model timesteps). Results are presented for the fully forced simulation, nudged simulations with $\tau_x$ of 4, 5 and 6 hours, and for the lower and upper quartile of the 50 perturbed simulations.

|  | Fully forced | $\tau_x = 4\,\text{h}$ | $\tau_x = 5\,\text{h}$ | $\tau_x = 6\,\text{h}$ | Pert. Q1 | Pert. Q3 |
|---|---|---|---|---|---|---|
| Inversion found (%) | 78.8 | 78.8 | 78.8 | 78.8 | 78.8 | 78.8 |
| Mean $z_{inv}$ mismatch (m) | 83.7 | 126.8 | 134.9 | 133.8 | 133.8 | 143.3 |
| St. dev. of $z_{inv}$ mismatch (m) | 64.2 | 101.7 | 119.5 | 122.0 | 117.5 | 125.0 |

Note: the percentage of the time the inversion is successfully reconstructed remains constant across the simulations because the stability criterion used depends also on large scale subsidence, which is unaffected by $\tau_x$ or perturbations in temperature and humidity, and in the case of the EPIC SCM experiment proves to be the limiting factor.

[Figure]

[Figure]

Figure A.1: Reconstructed inversion height in modified EPIC SCM experiments. Top: with different nudging relaxation timescales $\tau_x$; bottom: with $\tau_x = 5\,\mathrm{h}$ and perturbed initial conditions (IC).

---

## Author Response (AR3)

**Author Response to Reviews of**

**Vertical grid refinement for stratocumulus clouds in the radiation scheme of a global climate model**

Paolo Pelucchi, David Neubauer, and Ulrike Lohmann
*Geoscientific Model Development,* `doi:10.5194/gmd-2020-384`
* * *
RC: *Reviewer Comment*,     AR: *Author Response*,     ☐ Manuscript text

**1. Reviewer Comment #1**

RC: *In this paper, two related approaches to improve biases in stratocumulus cloud cover in ECHAM-HAM are presented. They both rely on a vertical grid refinement step. This allows the determination of the inversion level and then a remapping step, where the two methods differ. The updated vertical profiles are used in the radiation calculation. This is an interesting study. The results are basically negative, but are presented well. Given the nature of the results, I do not think it is incumbent upon the authors nor reviewers to try to "fix" these schemes or to subject the schemes to much additional scrutiny in this paper. The text identifies some of the difficulties with the approach. In my estimation, these boil down to (1) adjusting the cloud cover just for the radiation calculation does not seem to be enough to push the model toward a more realistic climate (in terms of stratocumulus cover) and (2) there are probably too many concessions in only applying the vertical refinement to the gridscale clouds to make it worth the effort for minimal improvements to the climate. I think only minor revisions are need for this paper.*

AR: *Thank you for your feedback. Please find our responses to your general points and specific comments below.*

**General points**

RC: *There is an opportunity here to make the work more broadly applicable by (1) making some comparison to other global models that might use similar approaches and (2) possibly making more assertive statements about what are likely to be productive paths toward improved stratocumulus representations in global models. On these two general points, I'll just add a couple thoughts.*

**Point (1)**

RC: *I noted in my specific comments below a connection two two other models. First is the NCAR CAM5 that used the UW moist turbulence and shallow convection schemes, which I think are based on Grenier and Bretherton. It would be interesting to know whether the results here could be related to results with that model (where the ambiguous layer must be used for determining turbulent mixing and possibly cloud cover). The other is the UCLA model that used a mixed layer model for their boundary layer scheme; that model isn't really relevant any more, but there are some shared assumptions with the scheme used here, so I wondered if there was any merit in making that connection?*

AR: *Thank you for indicating these two models to us. The approaches used in these studies to model the*

*stratocumulus-capped boundary layer do have a connection to our method, and the results obtained better inform the discussion around our own results. We have added a few sentences presenting these studies in our Introduction section; we further reference them in the context of our discussion of recommendations for future work.*

**Point (2)**

RC: ***On the second point, I think the conclusions here could be expanded a little bit. In particular, I wonder whether any recommendations could be made beyond the possibility of also applying the refined grid to the microphysics. Below I note some skepticism about that path, as it would seem to just lead to wanting to apply the refined grid to the rest of the physics too. One could also ask whether there is value in trying to better match the physics and dynamics by better including the inversion reconstruction in the vertical advection? Finally, and maybe most simply, how much benefit would there be to just increasing the vertical resolution in the boundary layer versus trying to reconstruct the smaller-scale structure?***

AR: *We agree this discussion deserves more consideration. We have expanded our Conclusion section with a discussion of potential advantages and disadvantages of applying the* invgrid *grid refinement scheme to other routines, of having a variable model level corresponding to the boundary layer top, and of using fixed grid levels but at higher resolution in the boundary layer.*

**Specific comments**

**1.1. Around line 105**

RC: ***The "invgrid" with the squished clouds seems like it would be problematic from the outset. Since the optical depth of the clouds is directly related to the LWP, when the cloud volume is conserved but made geometrically thinner but broader, the cloud fraction goes up, but the LWP would be reduced, correct? Seems like that could end up radiatively warming (via shortwave) the subcloud layer (a small effect) and the surface (potentially large effect in coupled settings, and would work against the goal of increasing cloud cover).***

AR: *The liquid water path (LWP) of the grid-layer is conserved: the layer thickness is reduced, but the liquid water mixing ratio is proportionally increased. The in-cloud LWP, which is what is used for cloudy-sky radiative calculations, is indeed reduced, by the same proportion that the cloud fraction is increased (we have corrected this in the manuscript). This is not problematic, as it is simply a more realistic representation of the situation. Also, the radiative flux calculation is linear in the cloud fraction but non-linear in the LWP (equivalently cloud optical depth). Therefore, there can be a difference in radiative fluxes by applying the* invgrid *scheme.*

*To better illustrate this point, we performed a simple radiative flux calculation. We consider only the shortwave (SW) for simplicity, as it is the dominant factor. In ECHAM-HAM, the radiative flux through a column grid-box observed at the top of the atmosphere (TOA) is:*

$$F_{allsky} = (1 - b) \, F_{clear} + b \, F_{cloud} \tag{1}$$

*where $b$ denotes the layer cloud cover. The SW cloud radiative effect (CRE) is defined as:*

$$CRE_{SW} = F_{allsky} - F_{clear} \tag{2}$$

We calculate the clear-sky and cloudy shortwave fluxes using radiative transfer equations from **corti_peter_2009**:

$$F_{clear}^{SW} \approx I_0 \left( 1 - r - tt^{'} \alpha \right) \tag{3}$$

$$F_{cloud}^{SW} \approx I_0 \left( 1 - r - tt^{'} \alpha - (1 - \alpha) tt^{'} \frac{R_c - \alpha R_c^{'}}{1 - \alpha R_c^{'}} \right) \tag{4}$$

where $I_0$ is the incoming solar flux, $r$ is the atmospheric reflectivity, $tt^{'}$ is the product of downward and upward atmospheric transmittance, $\alpha$ is the surface albedo, and $R_c$ and $R_c^{'}$ are the cloud reflectance for incoming and outgoing radiation respectively, calculated as

$$R_c \approx \frac{\tau/\zeta}{\gamma + \tau/\zeta} \tag{5}$$

$$R_c^{'} \approx \frac{2\tau}{\gamma + 2\tau} \tag{6}$$

where $\tau$ is the cloud optical depth, $\gamma = 1/(1 - g)$ and $g$ is the asymmetry factor, and $\zeta$ is the cosine of the solar zenith angle (SZA). In our calculations, the values used for the parameters are $r = 0.15$, $tt^{'} = 0.73$ and $\gamma = 0.77$ (**corti_peter_2009**), $\alpha = 0.05$ for the ocean, $SZA = 45°$ and $I_0 = 1360 \, \text{W m}^{-2}$ (solar constant).

We use as an example $\tau = 12$ and $b = 0.6$. If the layer thickness $Z$ is reduced by one third with invgrid ($Z^{ig}/Z = 2/3$), on the refined grid (denoted with superscript ig) we obtain $b^{ig} = b \cdot Z/Z^{ig} = 0.9$ and $\tau^{ig} = \tau \cdot b/b^{ig} = 8$. The resulting all-sky fluxes and SW CRE are

$$F_{allsky} \approx 724 \, \text{W m}^{-2} \tag{7}$$

$$F_{allsky}^{ig} \approx 613 \, \text{W m}^{-2} \tag{8}$$

$$CRE_{SW} \approx -382 \, \text{W m}^{-2} \tag{9}$$

$$CRE_{SW}^{ig} \approx -494 \, \text{W m}^{-2} \tag{10}$$

i.e. the effect of cloud squeezing is a reduction in the shortwave radiative flux reaching the surface.

To give a more general picture, we also investigated the influence of the initial layer cloud cover and of the thickness reduction ratio on the radiative effect of invgrid. Figure 1.1 shows the difference between new and old grid SW radiative flux.
Applying SC-VOLUME can be seen to always have negative (or at most null) SW radiative effect. From the results presented in Table 3 of the manuscript, on average the affected layers in stratocumulus regions have initial $b = 0.75$ and $Z^{ig}/Z = 0.95$. In these conditions the SW radiative effect is relatively small, around $-20 \, \text{W m}^{-2}$.

**1.2. Around line 130**

**RC:** *The column detection method is fine, but is this the complete description? Is there an ocean mask or a latitude limiter applied? Otherwise, it seems like non-subtropical-stratocumulus would be selected often (at high latitudes, for example). Oh, I see that later in the results, the changes outside the subtropics are noted. Was there any attempt to more directly limit the application to stratocumulus?*

AR: *We did not explicitly limit the regions of applicability of the method. The criterion used to select the columns in which to apply the method is based only on the low-tropospheric stability as a measure of the inversion strength. We chose to keep the criterion general and purely physics-based because, as the model struggles to represent sharp inversions in general, we believe that applying the method can be beneficial in any column with this problem, and not only for stratocumulus.*

[Figure]

Figure 1.1: Change in SW radiative flux due to application of SC-VOLUME cloud squeezing for a cloud with initial $\tau = 12$, initial $b$ as on the y-axis and a thickness reduction ratio as on the x-axis.

**1.3. Line 160**

**RC:** *A small notational thing, I always reserve q for specific humidity while r is used for mass mixing ratio. It's a convention that is known, but not always followed. I just want to confirm that it is mass mixing ratio that is being used (mass_water / mass_dry_air) and not specific humidity (mass_water / (mass_dry_air + Sum_i(mass_water_i))).*

AR: *Thank you for pointing this out. We are indeed using mass mixing ratio, and accordingly changed the notation to $r$ in the manuscript.*

**1.4. Line 200**

**RC:** *In climatologies stratocumulus will be thicker, but instantaneously, wouldn't we expect to often find much thinner layers?*

AR: *The "climatology" we were referring to from* **wood_2012** *is actually compilation of instantaneous thicknesses, rather than climatological thicknesses - we have corrected this in the manuscript. We introduced the $50\,\mathrm{m}$ threshold to avoid situations in which the reconstructed inversion would be found and/or get stuck on a half-level, and so the cloud would be almost infinitesimally thin, in order to avoid numerical problems and non-physical values.*

**1.5. Line 249**

**RC:** *I don't think that "heat content H" is an appropriate thermodynamic description for Equation 13; isn't this more correctly called the enthalpy?*

AR: *We agree that 'heat content' is not the appropriate term. By definition 'enthalpy' would also include the pressure-volume product, not present in our equation. We have hence changed it to 'internal energy', denoted by $U$.*

**1.6.   Sec. 2.2.3 (grid refinement)**

**RC:** *I'm interested to know whether regridding the aerosol fields was considered? I don't remember much about how ECHAM-HAM does aerosol, but I assume that there are a number of species that are advected and interact with the clouds and radiation. In some stratocumulus regimes, for example over the southeast Atlantic during biomass burning over central Africa, there are important aerosol direct (and possibly indirect) effects that influence the cloud/boundary layer structure. This seems like it could be problematic for this approach, since the aerosol will interact with clouds separate from the radiation, so it would not be obvious how to regrid the aerosol. If the aerosol is left homogeneously distributed in the grid cell, it could alter the radiative forcing in the column.*

AR:  *Aerosols are not regridded. The effect of aerosols on the clouds and boundary layer are still modelled normally, e.g. in the microphysics routine, where no grid refinement takes place. In the radiation routine, as they are left untouched in the affected layers, the aerosols' radiative forcing may indeed be slightly altered. However, we believe that the aerosol effects are of secondary importance to the effects due to change in cloud cover and cloud condensate. As we found these to be small, the aerosol radiative change is most likely negligible. Of course, for consistency the regridding of aerosols should be considered. We have added references to this point in Section 2.2.3 and in the outlook.*

**1.7.   Sec. 2.2.3 (grid refinement)**

**RC:** *The other question I had was whether rain or snow are radiatively active in the model. If so, it seems like they would need to be regridded as well – for example, to avoid the situation where drizzle is falling into the stratocumulus that made it (!).*

AR:  *Precipitation is not radiatively active in the model, so luckily, such situations do not arise.*

**1.8.**

**RC:** *Another question about the scheme itself is what it looks like for stratocumulus that are multiple grid levels deep? I assume this occurs frequently (it does in other models with similar resolution). It would seem like this would impose a structure that would be more like cumulus rising into stratocumulus in some circumstances. Or maybe I missed a detail, and there is some adjustment to the lower cloud layer, too? Later in the paper this is shown a little bit (Figure 5), and is kind of addressed in the discussion of the difference between the VOLUME and SUND schemes, but not completely. In the AMIP runs, I would expect multi-level clouds to occur frequently, and I'm still not sure if anything is done for the lower cloud layer in the case when the inversion level pushes the cloud top down into a level (rather than popping it up to the next level as in Fig 5b and c).*

AR:  *The scheme adjusts only the location of the cloud top by pushing the half-level atop of the uppermost cloud layer downwards, and therefore the lower layers of a multi-layer cloud are not affected. Given the restrictions for ambiguous layer choice we implemented (Sec. 2.2.1), it would not be possible to have a case where the inversion is reconstructed in a lower cloud layer leaving an unaffected cloud layer above. To clarify this point, we have added a sentence to explicitly address the treatment of multi-level clouds in Sec. 2.2.3.*
*As a related but separate issue, we considered adjusting the cloud base and attempted doing so by shifting the half-level under the lowermost cloud layer to the lifting condensation level, however we abandoned this idea as our reconstruction of the LCL was not successful (Sec. 3.1).*

**1.9. L330 / Fig 3**

**RC:** *The failure of the LCL diagnostic is interesting (although probably secondary to the main topic). It would informative to include in Figure 3 and indication of where "cloud base" is in the actual model. That is, mark the bottom of the level with nonnegligible liquid water. This seems to be indicated in Fig 5, so maybe it isn't worth adding to Figure 3.*

**AR:** *We have added an indication in the manuscript to point to the model cloud base and top shown in Figure 5.*

**1.10.**

**RC:** *Also, a 6-day SCM run isn't very convincing in terms of the success of the inversion reconstruction. Were other cases like DYCOMS-II or ASTEX also investigated? At L334, the comparison with previous results is noted. It is hard to have much confidence in this improvement based on what is shown. Another option would be to re-run the EPIC case a bunch of times with perturbed initial conditions (as in Hack & Pedretti 2000) to get a better sense of the statistical properties of the inversion reconstruction.*

**AR:** *We did the initial testing of the scheme in SCM with an idealised situation (no diurnal cycle) from the CGILS project (**zhang_etal_2013**), but this is not shown. We did not investigate other cases due to time constraints and lack of appropriate forcing files adapted to the version of ECHAM-HAM used.*

*Re-running EPIC with perturbed initial condition is an interesting idea, thank you for the suggestion. In our EPIC SCM simulations, the measured values of the vertical temperature and humidity profiles were used in the model at every timestep ('fully forced'). This allowed us to focus on analysing the performance of the inversion reconstruction scheme while being minimally biased by the SCM's ability to accurately reproduce a situation from a forcing. With this setup, a perturbation of the initial conditions (as in **hack_pedretti_2000**) would dissipate in a few timesteps. Therefore, for the perturbation experiments, we weakly nudged the SCM instead, with a relaxation timescale $\tau_x$ of 5 hours for temperature and humidity (see Eq. 25 in **lohmann_etal_1999**). The initial temperature and absolute humidity fields were perturbed following **hack_pedretti_2000**, i.e. for the temperature an normal additive perturbation with standard deviation of $0.5\,\mathrm{K}$ and absolutely bounded by $0.9\,\mathrm{K}$, and for the absolute humidity a multiplicative perturbation such that the standard deviation is $0.5\,\mathrm{g\,kg^{-1}}$ in the boundary layer and absolutely bounded by $6\,\%$. The experiment was run 50 times with different perturbed initial conditions. With this weak nudging setup, perturbations in the initial conditions can lead to differences in the reconstructed inversion height throughout the duration of the simulation. The new results are shown in Appendix A, Fig. A.1 and Table 1.*
*Compared to the previous fully forced simulation, the results appear less accurate at times, with more frequent sudden "jumps", particularly around the third day. However, the reconstructed inversion height is well in line with the observed cloud top during the first two and last two days. When the initial conditions are perturbed, the results for the inversion height deviate from the unperturbed simulation mostly during the first day, with only a few also deviating around days 4 and 5. Overall, despite the evolution being much more weakly forced, the reconstructed inversion height remains mostly consistent. This suggests that the inversion reconstruction method is robust.*

**1.11.**

**RC:** *I think Figure 4 should also include the radiosondes from EPIC that are mentioned previously. Looks like the data is available here:* `https://atmos.washington.edu/~breth/EPIC/EPIC2001_Sc_ID/sc_integ_data_fr.htm`

**AR:** *Thank you for the suggestion. We have added the measured vertical profiles to Figure 4 in the manuscript -*

*see Figure 1.2. We also show the same timestep in the nudged simulation mentioned in the previous point, to illustrate what the effect of the reconstruction might be in a more weakly forced situation where the profile was not already set to the measured one. The chosen timestep turned out not to be ideal to demonstrate the effect, as the real inversion already fell on a model level. However, it shows (especially in the nudged simulation) how even in such situations the grid refinement scheme can improve the shape of the profile.*

**1.12. Around L465**

**RC:** *This is a key conclusion of the paper, I think. If we think of this vertical regridding scheme as an attempt at some kind of "dynamic bias correction" to cloud cover, it doesn't really work. The initial cloud formation mechanisms are flawed, so a scheme that would just try to boost the cloud cover in the radiation is extremely limited in utility.*

**AR:** *Thank you, we agree. We have further highlighted this point - about ECHAM-HAM's lack of stratocumulus clouds in the first place - in the conclusion. We would also like to note however that improving the cloud cover in the radiation routine has worked in other models to improve other fields, e.g.* **boutle_morcrette_2010**.

**1.13. L502-4**

**RC:** *I agree that one might expect better performance by also applying the regridding to the the microphysics. That would be like an improved version of the SCSUND scheme that would deal with phase partitioning and drop numbers better (and aerosol?). I would suggest that approach would also come up short, and that the argument then would be that the turbulent mixing isn't represented correctly because it doesn't know the correct "mixing height" because it is acting (probably, depends on the scheme) on full model layers. So if the microphysics and radiation were adjusted, the recommendation might be to extend the adjustment to the turbulence, too. At that point, the Grenier and Bretherton (and then Bretherton and Park / Park and Bretherton) schemes would seem like an attractive solution, harmonizing the shallow convection, gridscale cloud physics, and turbulence; the microphysics and radiation then get to come along for the ride, but would depend a bit on the implementation. I don't think that in NCAR-CAM5 (which uses the Bretherton/Parks schemes) the radiation has any information about the inversion height.*

**AR:** *This is a valid point. However, we would not exclude a priori that an implementation in the radiation, cloud cover and microphysics routine could work, but it is worth mentioning that even then performance could be unsatisfactory because the information about the inversion is not being used in the vertical mixing scheme. In that case, a full PBL parametrisation scheme such as* **grenier_bretherton_2001** *may be preferable. We have added a discussion of these points in our manuscript, also in the context of our response to general point (2) above.*

*In the past an attempt was made to implement a scheme similar to* invgrid *(but where two new variable levels were added) fully interactively in ECHAM-HAM (see* **siegenthaler_2010**) *but the results were unsuccessful due to unspecified numerical problems. A study by* **boutle_morcrette_2010** *had obtained an improvement in global climate fields while implementing a simple stratocumulus cloud cover parametrisation only in the radiation scheme. This had encouraged us to investigate whether similar results could be obtained in ECHAM-HAM, however in our case we found the issue of the cloud layer mismatch, discussed in Sec. 3.3.1, particularly problematic for our approach.*

**EPIC SCM (fully forced)**

**EPIC SCM (nudged, $\tau_x$ = 5 h)**

Figure 1.2: Vertical profiles at the end of the fifth day (timestep 471) of the EPIC SCM experiments. Top: experiment with fully forced profiles; bottom: nudged experiment with 5-hour relaxation timescale.

**1.14.**

**RC:** *Another model that I thought about while reading this paper was the old UCLA GCM. The relevant idea there was that they used a well-mixed layer assumption to determine their lowest model level's height, which was synonymous with the "boundary layer". They did a relatively good job with stratocumulus because they had a level interface that was naturally at the inversion. (Now, where the mixed layer assumption didn't work well raised other important errors, but for stratocumulus it worked pretty well.) See Suarez et al. (1983) and Randall and Suarez (1984).*

**AR:** *Thank you for the references. We have added a few sentences referencing and discussing this model to our manuscript, as stated in our response to general point (1) above.*

**2. Reviewer Comment #2**

**RC:** *This paper describes the application of grid-refinement techniques to improve the cloud cover under inversions as seen by the radiation scheme in the ECHAM-HAM model. Ultimately, the attempts are unsuccessful in improving the mean model climate, which is somewhat sad as the paper is very well written. It leaves me quite uncertain what to suggest. On one hand, I'm supportive of publishing a study like this, as it is useful to the community to know what has been done, and that (in this case) it doesn't really work. On the other, I'm unsure that the paper contains enough new material to be published. In particular:*

  - *The method of grid refinement is not new, it is simply an application of an already published study (Grenier & Bretherton 2001).*

  - *The idea of giving the radiation scheme the spatial area of cloud seen rather than the volume fraction is not new, but has been discussed by several previous studies (most recently Boutle & Morcrette 2010).*

  - *Applying grid-refinement techniques to improve cloud cover has been more successfully implemented in other models, and so the application of this to a full GCM is not new. Further to this, the application in other models (e.g. Boutle & Morcrette 2010) has applied the technique to cloud variables throughout the model, rather than just to those seen by radiation. Therefore the previous studies on the topic seem to offer a more complete and consistent solution to the problem, and possibly unsurprisingly, have been more successful in demonstrating model improvements.*

*Therefore I'm struggling to see really what the new results being presented here are.*

**AR:** *Thank you for your feedback. We think that our study presents new ideas and results in the method used for obtaining the new-grid cloud cover in SC-VOLUME and especially in the insights about ECHAM-HAM's representation of stratocumulus clouds obtained from our analysis of **why** the proposed approaches had such little effect. As you mentioned, we used the inversion reconstruction method by **grenier_bretherton_2001** to find the location of the inversion and model the sub-grid thermodynamic profiles. However, the 'cloud squeezing' method applied to recalculate the new-grid cloud cover in SC-VOLUME, based on the simple and physically-motivated idea of using the inversion as the cloud top and conserving the cloud volume to obtain the real simulated horizontal cloud fraction, is a novel idea to our knowledge. In **boutle_morcrette_2010**, for example, the new horizontal cloud fraction used is the maximum value obtained from estimating the volume fraction on the three sub-levels onto which the inversion is sharpened by extrapolation. Additionally, in **boutle_morcrette_2010**, the new cloud cover was used only in the radiation routine ("The interpolation/*

extrapolation is only used to generate a value of $C_a$ to pass to the radiation scheme and for use in diagnostic outputs; it is not directly communicated to other parts of the model.", **boutle_morcrette_2010**, *Sec. 2.2), but improvements in other climatological fields were observed thanks to feedbacks. We had initially hoped that an implementation of the scheme limited to the radiation routine would be sufficient or beneficial in our model as well. Even though this turned out not to be the case, we believe that our subsequent investigation into the reasons behind the scheme's lack of success revealed interesting new insights about ECHAM-HAM's stratocumulus bias that could also be relevant for other models; most notably the cloud-inversion layer mismatch and the insufficient occurrence of stratocumulus clouds in the first place.*

**2.1. Suggestion 1.**

RC: *The best suggestion I can offer is to try the experiment applying SC-SUND to all cloud, not just that seen by the radiation. This would be consistent with how previous studies have applied similar techniques and demonstrated improvement. It would seem that you've done all the hard work in coding up the new scheme, and therefore linking it in to the main cloud water/fraction variables is a trivial extra step. This would (hopefully) not only allow you to show a model improvement that ECHAM-HAM developers/users would be interested in, but also allow discussion of why only applying the scheme to the radiation does not work.*

AR: *We think that applying the cloud cover calculated by the SC-SUND scheme for the refined grid elsewhere in the model would be problematic, as it would be inconsistent with the representations in the respective routines. For example, the microphysics routine is written with an interpretation of the cloud cover as a volume fraction, and hence using the SC-SUND cloud cover without also accordingly reducing the vertical resolution would result in an increase in cloud volume inconsistent with the microphysical calculations previously carried out. On the other hand, applying the grid refinement to other routines is a major undertaking and would require rewriting large parts of them. This is outside of the scope of this study, which was an attempt at investigating whether improving only the radiatively active cloud cover could improve it in the model in general.*

**2.2. Suggestion 2.**

RC: *I feel discussion of this point is somewhat lacking in the current paper. The expectation is clearly that this is the most important term in the cloud budget, and therefore should be sufficient - so why isn't it? It looks from Fig 6 that the increase in cloud from SC-SUND (e) is almost comparable to the bias in main model cloud (b). So is having improved radiative fluxes in these regions (I assume they are improved - this is something else that could be shown and discussed in the paper) not feeding back onto the inversion structure in a way that allows the cloud to form properly there? Or is the model vertical grid so coarse and inadequate that there is no hope of ever forming cloud correctly there? Both of these would clearly motivate diagnosing the full model cloud quantities using SC-SUND, as this will compensate for the poor vertical resolution, but also allow further improvements to the radiative fluxes and inversion structure, feeding back onto the cloud properties.*

AR: *We briefly discussed the reasons for the low radiative effect of SC-SUND in Sec. 3.3.3, but we agree that this point is deserving of more examination. Our proposed explanation is that the clouds newly formed with SC-SUND have too-low water condensate, because their water content is derived from the original grid where no cloud was present, so that they cannot produce a significant radiative difference.*

*To further motivate this explanation, we looked at the difference in SW cloud radiative effect (CRE) between the REF and SC-SUND simulations, which can give us some information about whether the new clouds formed in SC-SUND are radiatively different from the 'original' ones. As the CRE is the difference between all-sky and clear sky radiative flux, its magnitude can change based on changes in cloud cover, cloud occurrence*

*frequency or cloud optical thickness. If the mean cloud optical thickness remains the same, an increase in mean cloud cover in stratocumulus (Sc) regions as observed by the radiation routine in SC-SUND (both cloud cover and cloud frequency increased) should result in a more negative CRE in those regions. The following plot (Figure 2.1) shows the difference in mean SW CRE between the REF and SC-SUND simulations. Stippling shows regions where the difference is statistically significant at the 95% significance level.*

[Figure]

Figure 2.1: SW CRE difference between the REF and SC-SUND simulations.

*In Sc regions, where an important increase in mean cloud cover took place, the difference is still not significant. This can only be explained if the mean optical thickness of clouds in SC-SUND was much lower, compensating in the radiative effect the increase in cloud cover and occurrence. This provides an explanation as to why application of SC-SUND only in the radiation routine did not have the desired effect despite the large increase in cloud cover and occurrence - as suggested, the newly formed clouds are devoid of significant cloud condensate. If the new clouds were comparable to the usual clouds in terms of water condensate, a strong radiative change would be observed, leading to favourable feedbacks, like in SC-MAX. This motivates extending the* invgrid *grid refinement to the microphysics routine, which would then simulate a better representation of the cloud condensate to be passed the radiation routine.*

**2.3. Suggestion 3.**

**RC:** *My other suggestion would be to link the discussion to recent literature a bit more. Sundqvist-type cloud schemes that use a critical relative humidity are somewhat arcane and will always struggle around inversions due to the mixing of boundary-layer and free tropospheric air masses in a way that cannot be represented by a simple monomodal PDF and critical relative humidity. A (very) recent set of papers (van Weverberg et al. 2021a,b) has discussed this in detail, demonstrating that really the cloud properties here need to be considered as bimodal, and representing them otherwise probably places fundamental limits on how good the cloud can ever be near an inversion.*

AR: *Thank you very much for the recommendation. The idea proposed in* **vanweverberg_2021a***;* **vanweverberg_2021b** *could be a viable alternative approach to improve Sc cloud cover in layers that represent a mixture of dry and moist air, and where hence the Sundqvist approach underestimates it, so it is definitely worth discussing. We have added a reference to these studies and discussion in relation to ours to our manuscript.*

**3. Chief Editor Comment**

RC: *Dear authors,*
*in my role as Executive editor of GMD, I would like to bring to your attention our Editorial version 1.2:*
*https://www.geosci-model-dev.net/12/2215/2019/*
*This highlights some requirements of papers published in GMD, which is also available on the GMD website in the 'Manuscript Types' section:*
*http://www.geoscientific-model-development.net/submission/manuscript_types.html*
*In particular, please note that for your paper, the following requirements have not been met in the Discussions paper:*

AR: *Thank you for bringing these requirements to our attention. We apologise for not meeting them in the Discussion paper.*

**3.1.**

RC: *The main paper must give the model name and version number (or other unique identifier) in the title.*

AR: *We have changed the title of the manuscript to*

> *Vertical grid refinement for stratocumulus clouds in the radiation scheme of a~the global climate model ECHAM6.3-HAM2.3-P3*

**3.2.**

RC: *"Code must be published on a persistent public archive with a unique identifier for the exact model version described in the paper or uploaded to the supplement, unless this is impossible for reasons beyond the control of authors. All papers must include a section, at the end of the paper, entitled "Code availability". Here, either instructions for obtaining the code, or the reasons why the code is not available should be clearly stated. It is preferred for the code to be uploaded as a supplement or to be made available at a data repository with an associated DOI (digital object identifier) for the exact model version described in the paper. Alternatively, for established models, there may be an existing means of accessing the code through a particular system. In this case, there must exist a means of permanently accessing the precise model version described in the paper. In some cases, authors may prefer to put models on their own website, or to act as a point of contact for obtaining the code. Given the impermanence of websites and email addresses, this is not encouraged, and authors should consider improving the availability with a more permanent arrangement. Making code available through personal websites or via email contact to the authors is not sufficient. After the paper is accepted the model archive should be updated to include a link to the GMD paper."*

AR: *We have added a link to the SVN tag corresponding to the specific model version used to the* Code and data availability *section:*

*The ECHAM-HAMMOZ model is made freely available to the scientific community under the HAMMOZ Software License Agreement, which defines the conditions under which the model can be used.* *The specific version of the code used for this study is archived in the ECHAM-HAMMOZ SVN repository at* `/root/echam6-hammoz/tags/papers/2021/Pelucchi_et_al_GMDD`. *More information can be found at the HAMMOZ website (https://redmine.hammoz.ethz.ch/projects/hammoz, last access: 21 September 2020).*

*For the final paper a new SVN tag will be added and this link will be updated. The code can be accessed if the HAMMOZ License, which is free for research, is obtained.*

**RC:** **Please note, that even though the code is not freely available, the exact code version used for the publication needs to be archived. There please provide an identifyer or other means how the exact code version can be accessed.**
**Please add the name and version number of the model used (ECHAM-HAMMOZ) and the version number to the title of your manuscript.**
**Yours,**
**Astrid Kerkweg**

AR: *We hope that our modifications and additions will be considered in line with the requirements.*

**A.   Appendix: Modified EPIC SCM experiment**

Table 1: Results from the modified EPIC SCM experiments: percentage of the time the inversion is reconstructed, and mean and standard deviation of the mismatch (absolute difference) between the reconstructed inversion height $z_{inv}$ and the measured cloud top (linearly interpolated to all model timesteps). Results are presented for the fully forced simulation, nudged simulations with $\tau_x$ of 4, 5 and 6 hours, and for the lower and upper quartile of the 50 perturbed simulations.

|  | Fully forced | $\tau_x = 4\,\mathrm{h}$ | $\tau_x = 5\,\mathrm{h}$ | $\tau_x = 6\,\mathrm{h}$ | Pert. Q1 | Pert. Q3 |
|---|---|---|---|---|---|---|
| Inversion found (%) | 78.8 | 78.8 | 78.8 | 78.8 | 78.8 | 78.8 |
| Mean $z_{inv}$ mismatch (m) | 83.7 | 126.8 | 134.9 | 133.8 | 133.8 | 143.3 |
| St. dev. of $z_{inv}$ mismatch (m) | 64.2 | 101.7 | 119.5 | 122.0 | 117.5 | 125.0 |

Note: the percentage of the time the inversion is successfully reconstructed remains constant across the simulations because the stability criterion used depends also on large scale subsidence, which is unaffected by $\tau_x$ or perturbations in temperature and humidity, and in the case of the EPIC SCM experiment proves to be the limiting factor.

[Figure]

[Figure]

Figure A.1: Reconstructed inversion height in modified EPIC SCM experiments. Top: with different nudging relaxation timescales $\tau_x$; bottom: with $\tau_x = 5\,\mathrm{h}$ and perturbed initial conditions (IC).

**Author Response to Reviews of**

**Vertical grid refinement for stratocumulus clouds in the radiation scheme of the global climate model ECHAM6.3-HAM2.3-P3**

Paolo Pelucchi, David Neubauer, and Ulrike Lohmann
*Geoscientific Model Development,* `doi:10.5194/gmd-2020-384`
* * *
**RC:** *Reviewer Comment*,     AR: *Author Response*,     ☐ Manuscript text

**1. Report #1**

**RC:** *I thank the authors for their considered response to comments on the previous version of the manuscript. In my estimation, this manuscript has addressed the comments adequately and should be accepted for publication. As was mentioned in the previous reviews, the results are essentially negative, making it a little bit of a challenge to review (probably even more so to write). The authors have provided context for the results that include physical understanding in terms of the structure around the inversion and the limitations of Sundqvist-type schemes as well as relating to other models and other methods. There are several potential avenues for future work identified, including extending the grid refinement to the microphysics or adopting a different approach as in Weverberg. Although the approach used here is not novel, it documents a new implementation in this model, and the results differ from those shown with other models. This may not be the final word in terms of understanding why the grid refinement approach may not always work well, but I think it is a positive step toward improved understanding of inversion reconstruction approaches and their limitations.*

AR: *Thank you very much for your feedback and review.*

**2. Report #2**

**RC:** *Thanks to the authors for their efforts in revising the paper. The updated manuscript is significantly improved, and I can now see a path to publication. However, I have a few clarifications & queries arising from the responses:*

AR: *Thank you very much for your feedback and review. Please find our answers to your comments and questions below.*

**RC:** *I apologise for my slightly misleading comments about Boutle & Morcrette 2010. The discussion in there that I was thinking of was: "Cv and the condensate amount (qc ) are obtained from averages of Cv and qc on the sublevels, and are used when calculating the microphysical transfers that lead to the formation of precipitation (Wilson and Ballard, 1999)." i.e. when the UM is run with a diagnostic cloud scheme (similar to ECHAM-HAM), the sub-level interpolation method is used to create new (increased) values of*

*condensed water and volume cloud fraction, which are communicated to all parts of the model, not just the radiation. It was this experiment that I was suggesting with SC-SUND, i.e. in that case, a very similar thing is happening - you are diagnosing new (increased) condensed water values, which could be passed to all parts of the model, not just the microphysics. You would still be using the volume assumption for cloud fraction. I agree that in your application of SC-VOL and SC-MAX, where you only calculate the area cloud fraction, and do not alter the condensed water, it is only appropriate to pass this to the radiation (as with the prognostic implementation discussed in Boutle & Morcrette 2010 which you mention).*

*However, your answer to the next point makes me think that this probably wouldn't actually be that helpful, because it appears that the amount of additional cloud being created by SC-SUND is negligible. Do you understand why such a negligible amount of extra condensate is created with SC-SUND? The increase in fraction (up to 30%) seems quite significant, and presumably given the constrained link between fraction and condensate in the Sundqvist scheme, the increase in condensate is commensurate to what would otherwise be obtained in clouds of this fraction?*

**AR:** *In SC-SUND the water condensate is calculated as in SC-VOLUME (cf. Sec. 2.2.3, Eq. 11), i.e. potentially increased in the ambiguous layer but overall conserved. The Sundqvist cloud cover scheme is based only on relative humidity, i.e. the amount of water condensate has no effect. Therefore in SC-SUND the cloud cover is increased because of the higher RH but the amount of water condensate is not affected. In our schemes the water condensate amount is based on the pre-existing values and so is not automatically increased proportionally with the cloud cover increase. Normally the cloud microphysics scheme is run after the Sundqvist cloud cover scheme to calculate appropriate water condensate values. In our case, as previously mentioned, adapting the microphysics scheme to allow it to use the new grid and newly calculated values of cloud cover, $T$ and $q_v$ in the cloudy layer would be a major undertaking and was out of the scope of our study. This is what we would suggest as potential future work to improve the efficacy of SC-SUND.*

**RC:** *Which leads me to a further question/clarification. My understanding is that SC-SUND is only used to create new clouds where previously there were none? What if it were also used to increase the water content of pre-existing clouds? The inversion sharpening code could be used in exactly the same way to re-diagnose the water content of previously existing clouds. If, what you have discovered so far, is that clouds with fraction of 0-30% are radiatively unimportant due to small water contents, increases to the water content of previously existing clouds should stand a better chance of being radiatively important. Is it the case that the increase in water content with the Sundqvist scheme is quite sensitive to the cloud fraction, i.e. increasing fraction from 0-20% gives a smaller increase in water content than increasing from 40-60% would (for example)?*

**AR:** *In SC-SUND the cloud cover is also re-diagnosed with Sundqvist for layers which already contained a cloud. The water condensate content is not affected for neither pre-existing clouds nor new clouds (see our answer to your previous comment). We have found that running the Sundqvist scheme on the new grid results in the formation of a new cloud (i.e. new cloud cover > 0 where condensate is present) up to 30% of the time in Sc regions (Fig. 9b). However we have also seen that these new clouds are not radiatively important because the condensate amount is too low (Fig. 10), as indeed it is not appropriately re-calculated by the microphysics scheme to be in line with the new cloud cover. This is a limitation of SC-SUND that would be corrected by extending the new grid representation to the microphysics routine.*

**RC:** *I would still push the authors to investigate some combination of the above experiments, as I feel it could be useful in understanding what is going on. Certainly some further discussion on the behaviour of the Sundqvist scheme and why it does not produce much condensed water at low cloud fractions would be useful.*

AR: *The purpose of the Sundqvist scheme is only to diagnose cloud cover, based on RH; it is not meant to produce or alter condensed water (see our answer to your previous comments). We have added some discussion clarifying this.*

RC: **I also think that you could strengthen some of the discussion about the frequency of occurrence of stratocumulus clouds in the model (and perhaps include this in the abstract). I think this strengthens your work. The methods you propose are only really useful for targeting errors in "amount when present" (with the exception of SC-SUND). But it appears the main bias in the model is in "frequency of occurrence", and therefore it is possibly unsurprising that the proposed changes have limited benefit. It could be argued they would be much more successful in models with good frequency of occurrence and poor amount when present.**

AR: *Thank you for the suggestion. The main practical limitation of the method comes from the inversion layer/cloud layer mismatch rather than the stratocumulus occurrence frequency in general, but the framing of the method as targeting "amount when present" errors is useful. We have added the discussion of these points and also highlighted them in the abstract.*